# INFERRING CAPABILITIES FROM TASK PERFORMANCE WITH BAYESIAN TRIANGULATION

## ABSTRACT

As machine learning models become more general, we need to characterise their capabilities in richer, more interpretable ways that move beyond aggregated statistics on static benchmarks. We describe a method to infer the *cognitive profile* of a system from diverse experimental data. To do so, we introduce *measurement layouts* which model how task-instance features interact with system capabilities to explain performance. System capabilities can be estimated using Bayesian triangulation, inferring their value based on the performance of a model on tasks with different features. Our approach accurately recovers the cognitive profiles of hand-crafted behavioural agents, as well as estimating the cognitive profiles for deep reinforcement learning agents and human children in a virtual game environment. These cognitive profiles are significantly richer than aggregated benchmark statistics, summarising multiple distinct capabilities that explain behaviour, and are also significantly more predictive, accurately estimating performance on new, held-out tasks.

## 1 INTRODUCTION

What does success or failure tell us about a system's capabilities? In isolation, performance results are hard to interpret: most interesting tasks require several different capabilities to be successfully combined. For example, consider an agent exploring a 3D environment similar to that in Figure 1a, where in order for the agent to find the reward, it needs to understand that the reward still exists when occluded (known as object permanence). It must then remember where it is, and successfully navigate to it. If the agent fails on an instance of this task, the cause is unclear—is it a failure of object permanence, limited memory, or navigation skills? Without an answer to this question, it is difficult to predict performance for new tasks with different features. One way to approach this issue is to collect the performance of the agent on a benchmark of arbitrarily varied tasks (with and without occlusion, shorter or longer distances to the goal, etc.), and compute the proportion of successes across that set. This *aggregated statistic* approximates the expected probability of success on a new task instance, but it does so blind to its specific features. Moreover, this approach fails to *explain why* the agent passes or fails on a certain task.

Suppose, however, we also knew that the agent performs well on tasks that involve complex navigation, and the same for tasks involving memory. We could now infer, by triangulation, that the system's failure is likely a failure of object permanence. This is a robust inferential method in formal epistemology (Heesen et al., 2019) and causal (Bayesian) reasoning (Pearl, 2009).

This bottom-up inference is illustrated in green and red in Fig. 1b. We can use this information to draw conclusions about where the system is safe to deploy, and to direct efforts to improve it. We can also predict behaviour in new scenarios: for example, the agent will likely fail at a new task with high object permanence demands, as illustrated in blue in Fig. 1b.

This example demonstrates that we can leverage the fact that different tasks—and their constituent instances—have different demands to help us extract conclusions about system capabilities from patterns of task performance. However, these conclusions require an analytical approach that facilitates complex inferences. If we were to examine each task or instance in isolation or to simply aggregate performance across all tasks, this triangulation of abilities would not be possible. To address this, we propose a novel Bayesian approach to evaluation that enables simultaneous inferences

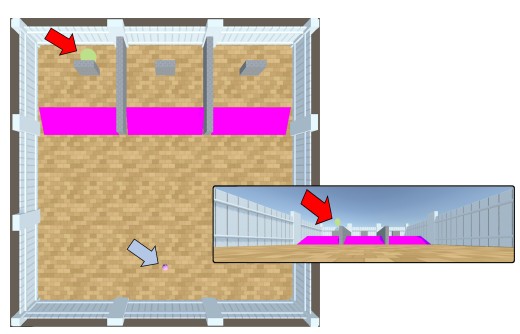

(a) A three-cup instance. The agent must (a) retain the object's existence when occluded, (b) remember which wall hides it, and (c) navigate around obstacles.

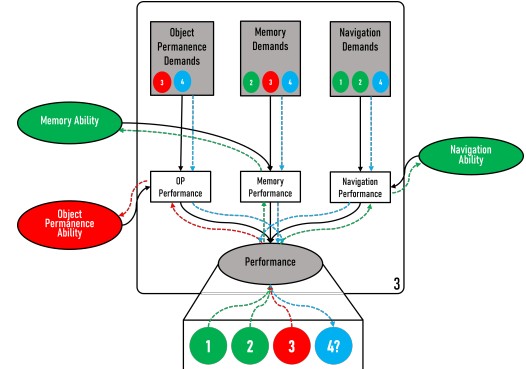

(b) Bayesian triangulation: bottom-up inference (green/red) yields a cognitive profile; top-down inference (blue) predicts performance on a new task.

Figure 1: (a) shows a representative task instance; (b) shows how Measurement Layouts infer and predict capabilities from instance-level results.

about multiple capabilities from patterns of task performance. This approach is underpinned by several psychometric principles that we elucidate in Appendix A.

In this paper we find an operational solution to these inferential challenges in the form of *Measurement Layouts*: specialised and semantically-rich hierarchical Bayesian networks. Using the inference power of the No U-Turn Sampler (Hoffman & Gelman, 2014) in PyMC (Salvatier et al., 2016a), a probabilistic programming engine, we can triangulate from performance and the task's demands to the capability and bias levels for each subject (their *cognitive profile*). Then, performance can be inferred downstream for new tasks. Not only are cognitive profiles much more explanatory of what causes failure or success, but we find that they are also much more predictive than the traditional approach of using aggregated benchmark statistics to predict performance.

Our approach is agnostic to the task or type of system being evaluated, so long as there are instance-level task demands that can be related to distinct capabilities. To showcase our method, we construct measurement layouts and cognitive profiles for agentic systems in two scenarios: a simple navigation task and a more complex object permanence task.

The paper is organised as follows: Section 2 introduces the measurement layout framework in general, Sections 3, 4, and 5 contain our three experimental settings. Section 6 provides general principles for building effective measurement layouts. Section 7 situates our work in the broader context ML evaluation, cognitive modelling, and latent variable/Bayesian modelling. Finally, Section 8 discusses the advantages and tradeoffs present in this framework.

## 2 MEASUREMENT LAYOUTS

Performance can be understood as a function of task *demands* and capability *levels*. The precise relation between the two depends on the task features and the tested capabilities. For instance, in a memory problem, the number of objects to remember in a particular task demands a memory capability level at least as high as the number of objects. However, the difference between task demands and capability levels is likely not sufficient to account for all variance in observed performance. Cognitive biases, such as a preference or aversion to apparently irrelevant features like colour, can also affect performance. Finally, AI systems can exhibit a lack of robustness caused by some unaccounted factors or noise. Cognitive profiles need to include all of these.

We define a **task characterisation** as the set of observable, usually constructed, meta-features $X$, expressing cognitive demands and other high-level properties of the task. For example, in an image classification problem, the level of blur of the image or the level of clutter of the background could constitute perceptual demands. Many other properties may also, in practice, affect performance.

Similarly, we characterise the **cognitive profile** of an AI system as a tuple $\langle C, B, R \rangle$. Where $C$, $B$, and $R$ are vectors of capability levels, bias, and robustness values. This tuple is simply to help the evaluator separate organise latent variables into conceptual groupings; within the Bayesian model, all latent variables are treated identically. Capability levels represent what the system can do, bias values represent some other preferences or limitations that may impact performance in a less monotonic way, and robustness levels account for reliability issues, unexplained or random effects (noise). When we say bias may impact performance in a less monotonic way, we mean that more capability generally only improves task performance, but with bias, performance may degrade from some ideal value (typically a bias value of 0).

We define a **measurement layout** as a directed acyclic graph that connects, through linking functions, the meta-features of a task characterisation with the cognitive profile of an agent, in order to predict observed performance. We implement measurement layouts as Hierarchical Bayesian Networks (HBN) (Murphy, 2023) with the following characteristics: Within the measurement layout we include a node for every meta-feature in every task instance and a node for each element of the system's cognitive profile. All of them are roots of the graph. A connection $A \rightarrow B$ denotes that $B$ is conditionally dependent on $A$. Linking functions are used to formally define relationships between dependent nodes. Nodes can encode meta-features, latent capabilities, and derived nodes such as observable performance or Intermediate Non-Observable Nodes (INON) for representing intermediate marginal performances. All three node types were seen in Fig. 1b. The connective structure can be enriched by domain knowledge determining which cognitive profile elements and meta-features are expected to affect the values of instance-level inference nodes. These relationships can be higher-order and complex, with certain INONs requiring complex combinations of many cognitive profile elements and meta-features. The need for complexity can arise from the many confounding behaviours that a well-designed benchmark needs to account for. By a linking function, we refer to any differentiable function that maps the values from the outputs of one (or multiple) node's probability distribution to the input of another. For example, it could be a sigmoid function of the difference between capabilities and demands. The functions can scale the incoming nodes, so that we adjust the ranges for capabilities, biases and noise. We give the specific equations for the linking functions (and node formulations) that we used in this work in Tables 11 and 3 in the appendix.

A key conceptual difference between our measurement layouts and typical uses for HBNs is the fact that the structure of the measurement layouts are built to capture hierarchical dependency relations between demands and capabilities, allowing us to capture domain knowledge; whereas typical HBNs encode information on hyper-priors about factors to measure. The linking functions between nodes are often non-linear, conceived from domain knowledge of the task and applying ideas from cognitive modelling. One of the key decisions is to determine how capabilities interact and the extent to which they are compensatory, being realised with additive or multiplicative expressions.

We use PyMC (Salvatier et al., 2016b) to implement and fit measurement layouts in practice using the No U-Turn Sampler (Hoffman & Gelman, 2014), although any probabilistic programming language can be used for this.

It's important to note that measurement layouts do not rely on a population of agents to fit. But rather make inferences using performance results of a single AI system to estimate capabilities of that AI system alone. This provides an advantage insofar as we are not reliant on reference populations of AI systems or humans.

For the Bayesian triangulation (backwards inference) process to accurately infer the capabilities, we require that the tasks are designed with evaluation in mind. Intuitively, triangulation requires two factors: 1) There are task instances where one demand is high and the others are low (control tasks for a specific demand). This allows us to see performance changes and assign "blame" to a particular capability and 2) Different demands are varied in different ways across instances— rather than always co-varying with high-demand / low-demand. If two demands are always co-varied in exactly the same pattern across tasks separating the reason for success / failure will be significantly more difficult. In the following sections, we have built the task instances with this systematic variation in mind. Additionally, it's important to keep in mind that the capabilities we are trying to evaluate are latent constructs. This means that they are *indirectly measurable* and as such, ground truth values are difficult to ascertain. The way we address this is two-fold. First, experiment two uses synthetic data where we can create known ground truth data. Second, one

metric for evaluating whether the measurement layouts effectively measure agent capabilities is analysing their *predictive validity*—whether they can be used to predict agent success on held out instance results.

# 3  EXPERIMENT 1: A SIMPLE NAVIGATION TASK

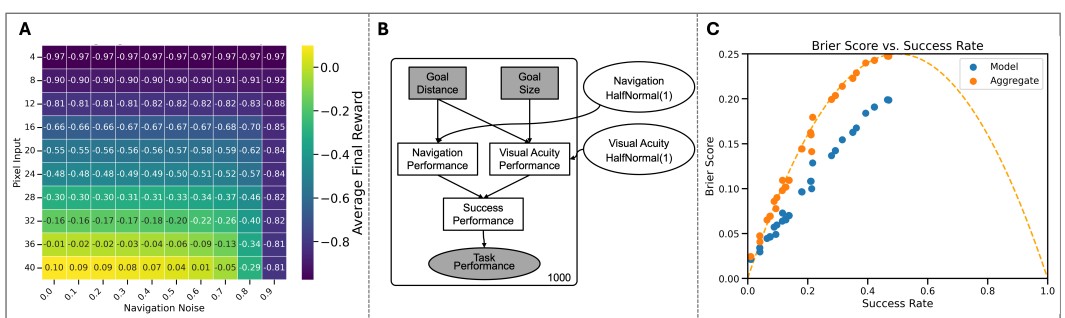

Figure 2: A: The average final score of each agent, varying pixel input size and navigational noise. B: A measurement layout relating two task demands and a capability to performance. C: The Brier Scores on the held-out test set of performances, as predicted by the average performance on the train set (Aggregate; Orange) and the measurement layout (Model; Blue), against the expected Brier Score on the test set (orange line). The measurement layout outperforms the aggregate for every agent. **Note**: No agent passed more than around 50% of the instances.

To demonstrate the power of the measurement layout approach, we design a simple experiment with agents with predefined capabilities. These agents interact with a three-dimensional embodied environment, Animal-AI (Beyret et al., 2019; Voudouris et al., 2025), and can move forwards, backwards, and turn; their goal is to obtain a green reward in each episode. They receive an $n \times n$ RGB image, and turn around until there is a green pixel directly ahead, and then they move towards it. We can vary the resolution of their image observations, directly manipulating their capability to see, i.e., their visual acuity. We can also vary the speed with which they move towards green rewards, by randomly forcing the agent to perform stationary "no-op" actions once they have located a green reward, directly manipulating their navigation capability. We generate 1000 agents with different visual acuities and navigation capabilities, ensuring full coverage of the capability space, and test them on 1000 tasks in Animal-AI consisting of green rewards of different sizes placed at different distances from the agent. The average final reward across these 1000 tasks is presented in Figure 2A, demonstrating the effect of manipulating the size of the observation image and the amount of navigational noise.

Figure 2B presents the measurement layout for this task, relating the goal distance and the goal size to two latent capabilities: visual acuity and navigation capability. We relate visual acuity to the quantity $\frac{distance}{size}$. Smaller, more distant rewards are harder, and therefore require higher visual acuity, than larger, closer rewards. Navigation capability is only related to goal distance. We use half-normal priors on these latent capabilities, reflecting the fact that there is a meaningful 0 value for both navigation capability and visual acuity, representing no capability. We fitted the measurement layout in Figure 2A using 4 chains, with 1000 warm up draws and 2000 sample draws to ensure convergence. Figure 2C shows how our measurement layout outperforms the aggregate as a predictor of success for all agents. Figure 3 shows that the inferred posterior estimates of the capability neatly correspond to the relative visual acuity and navigation capability of a sample of agents.

# 4  EXPERIMENT 2: AN OBJECT PERMANENCE TASK

Our second experiment explores a more complex set of capabilities, including Object Permanence. To ensure that we knew the ground truth abilities of the agents, we created a synthetic dataset by simulating the performance of 30 synthetic agents based on a diverse range of cognitive profiles. To achieve this, the authors split into two teams: Team A who designed the synthetic agents, and Team B who built the measurement layout without knowledge of the agents built by Team A. More

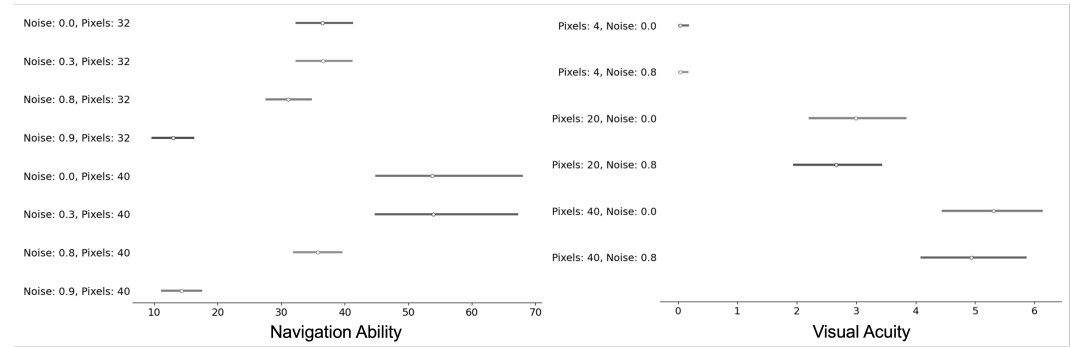

Figure 3: Abilities for a selected subset of vision agents, with means (points) and 95% Highest Density Intervals. Left: The navigation abilities for agents with 32×32 and 40×40 pixel inputs, for noise levels of 0.0, 0.3, 0.8, and 0.9. Differences in navigation are recovered given a pixel input size. Right: The visual acuities for agents with 4×4, 20×20, and 40×40 pixel inputs, for low (0.0) and high (0.8) navigational noise. Differences in visual acuity are recovered given a navigational noise level. Navigation ability is dominated by visual acuity because the agent can only navigate (with noise) to a goal that it can see.

details of the specific testbed and the synthetic agents can be found in Appendix C.2. We present the measurement layout for the object permanence tasks in Fig. 4. In the object permanence benchmark (Voudouris et al., 2022), there are additional challenges to navigation in the form of ramps, platforms, and lava. Subsequently, meta-features for the presence of these challenges, as well as nodes for the agent's capability (and instance-level performance) at handling them must be added to the measurement layout. In order to represent object permanence specifically, we add extra meta-features corresponding to key OP demands. A full mathematical description is given in Appendix D.2

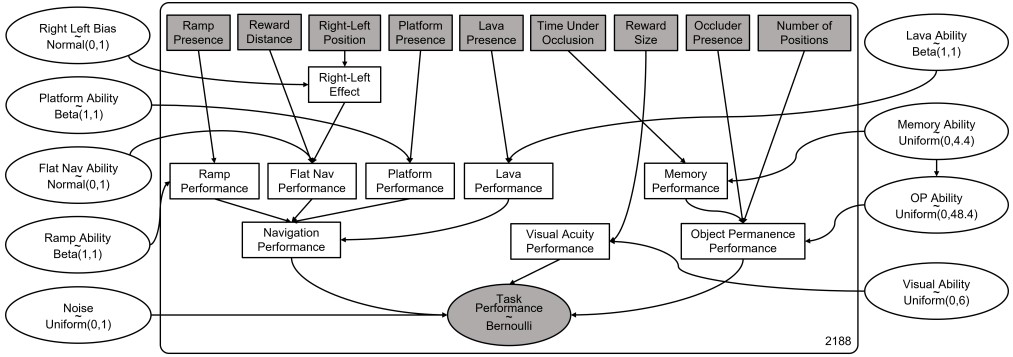

Figure 4: Visual representation of the measurement layout for the object permanence task.

As before, the cognitive profiles of agents were inferred with PyMC using 4 chains, 1000 warm-up samples, and 2000 sample draws, noting one divergences (agent 3) across all agents. Overall, our model was successful at inferring system capabilities from patterns of performance. We evaluated the model in two ways, as we see next.

**Qualitative Evaluation**  The first validation was a qualitative comparison of the cognitive profile descriptions generated by Team B with the criteria used to generate the agents. The measurement layout facilitated a correct description of the agent in around 75% of cases. Where agents had been designed to have a specific weakness ("Achilles Heel" agents), the model was highly successful at identifying the cause of failure. None of the Achilles Heel agents (6,9,11,13) showed degraded performance on Object Permanence when other factors caused failure on specific tasks (Figure 5b). Agents without Object Permanence (Agent 3) have a reduced score. Agent 5 passes all tests with high probability and is given a high OP score. However, each Achilles heel agent had its weakness

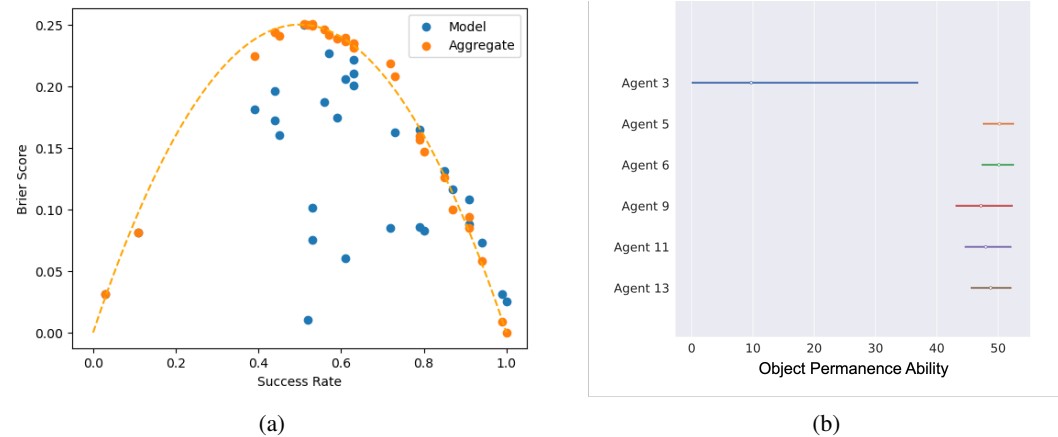

|     |     |
| :-: | :-: |
| (a) | (b) |

Figure 5: **a)** Brier score of model predictions against agent success rates within O-PIAAGETS. **b)** Posterior means and 95% HDI for a selection of interesting agents on Object Permanence capability.

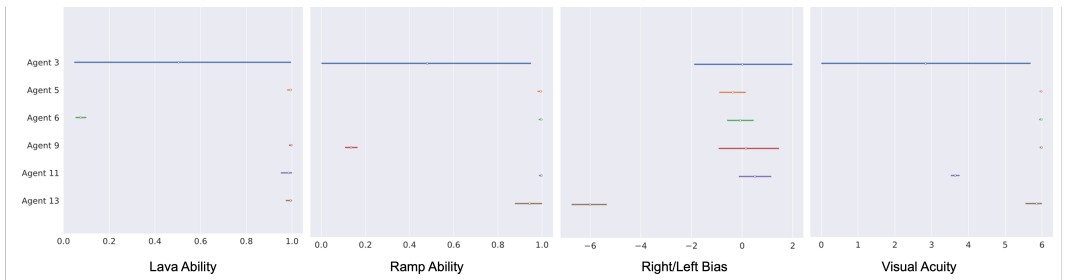

Figure 6: The posterior means and 95% Highest Density Intervals for a selection of interesting agent types on four capabilities: Lava, Ramps, Right-Left Bias, and Visual Acuity.

identified in the relevant capability (Figure 6). It was, however unable to spot "fraudsters" making use of strategies (such as "go to the last seen location of the reward") to mimic OP capability. Appendix E.1 contains a detailed breakdown of inferred parameters of the cognitive profiles of each agent, more information on the experimental set up, and the full qualitative analysis comparing inferred capabilities to the ground truth capabilities of the agents designed by Team A. E.2 contains a deeper analysis of the results, looking at different categories of agent in more detail, and E.3 contains the full forest plots for all agents and core capabilities.

**Predictive Evaluation**    For our second validation, we show that our model can be used for accurate prediction of agent success on unseen task instances. Taking instance meta-features and applying forward inference with our model yields a probability of success. We compare this to the probability predicted by aggregate success. A full analysis is available in Appendix E.3. However, broadly, we see that the model is consistently a better predictor of success (lower Brier score; Figure 5a) than relying on the aggregate measure. We see better prediction performance for agents with less certain success rates. These are the agents for which we are more interested in improving prediction.

## 5    EXPERIMENT 3: EXTENDING MEASUREMENT LAYOUTS TO REAL DATA

In our third experiment, we adapted the previous measurement layout to construct cognitive profiles for real agents solving object permanence tasks, using data from an independent study (Voudouris et al., 2024). We fitted cognitive profiles for deep reinforcement learning agents (Proximal Policy Optimization, PPO, Schulman et al. (2017); Dreamer-v3, Hafner et al. (2025)) on five different training curricula (Voudouris et al., 2024), as well as human children aged 4–7 years old, a simple heuristic agent similar to those described in Section 3, and a random action agent that moves stochastically in the environment. We use the random and heuristic agents and children to calibrate

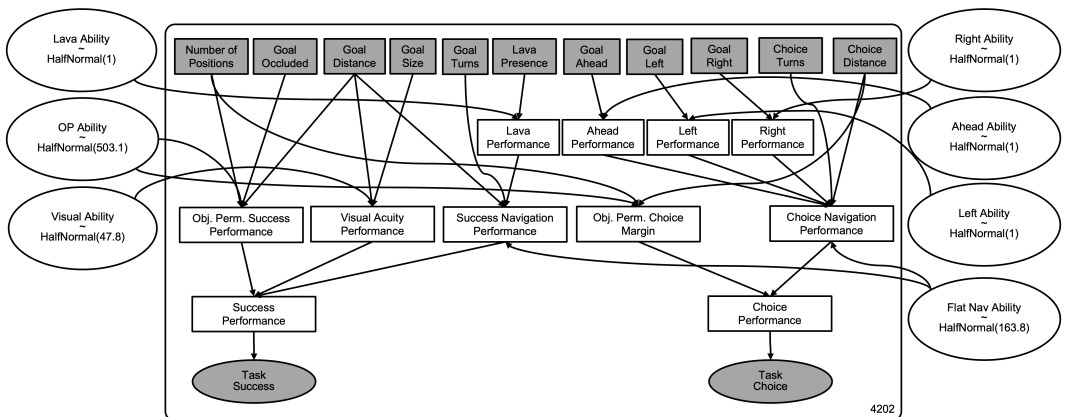

Figure 7: The updated measurement layout to accommodate the meta-features and observed variables collected in Voudouris et al. (2024).

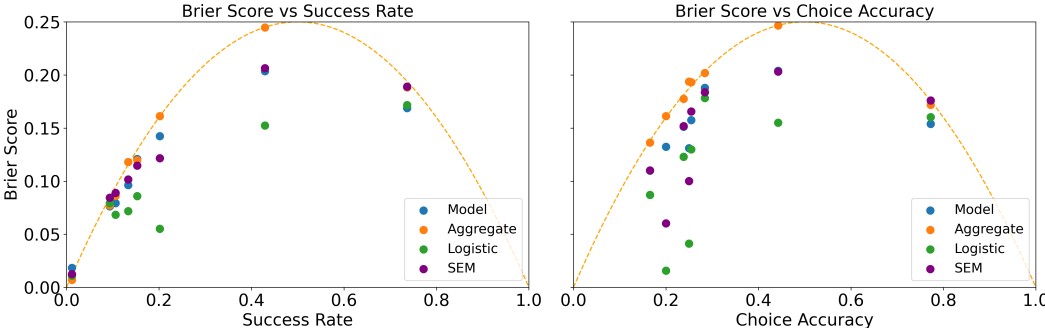

Figure 8: Brier Scores for Success (Left) and Choice (Right), compared to the aggregate predictor, a logistic regression, and a Structural Equation Model (SEM) with identical structure to our measurement layout.

our measurements against known baselines—they allow us to validate that our measurement results are coherent with the expected cognitive profiles of these known agents. This improves our trust in the measurements about the unknown PPO and Dreamer-v3 agents.

These behavioural data were more complex than those generated for Experiment 2: the agents were evaluated on whether they entered the same location as the occluded reward (choice response) as well as whether they ultimately obtained the reward (success response). We therefore generalised the measurement layout to incorporate this multivariate outcome. We also made some adjustments to the capabilities and meta-features to be inferred, to account for differences in the datasets. A detailed account of the updated measurement layout is given in Appendix D.3, but is also presented in Figure 16. We also fitted measurement layouts for children both with and without (ablated) a *visual acuity* capability to account for the lack of variance in the goal size meta-feature. Our measurement layouts were fitted to 10 deep reinforcement learning agents, 1 random agent, 1 heuristic agent, and the combined dataset of all instances from children aged 4–7 (n=1608; with and without visual acuity ablation). Four chains were used for each, with a 1000 sample warm-up and 2000 draws.

Figure 8 shows the Brier Scores for the measurement layouts on the held-out test set compared to using the average performance on the train set as the predictor. All measurement layouts are better than the Brier Score except for the Dreamer 4, Dreamer 5, PPO 4, and PPO 5 for the success response variable. This is likely because performance is so low for these agents that the aggregate measure is very predictive of overall performance. Noticeably, we are able to fit a predictive model of the behaviour of children, who were capable of completing the task and also had high levels of variability due to the between-subjects design.

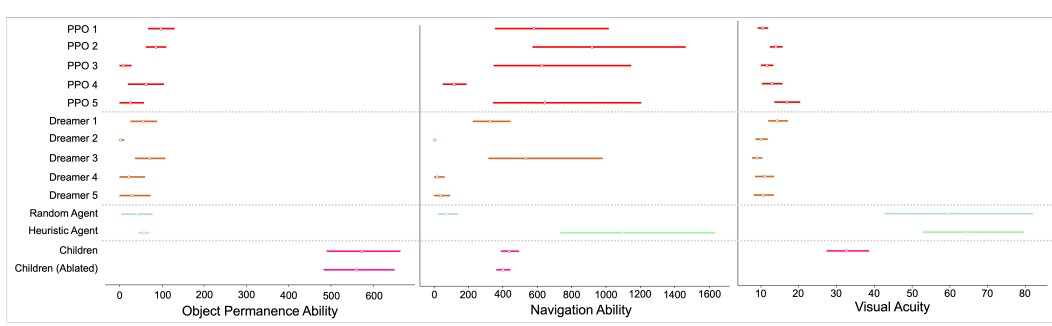

Figure 9: The posterior means and 95% Highest Density Intervals for 13 agent types on three capabilities: object permanence, navigation, and visual acuity.

We also compare the predictiveness of measurement layouts to two other predictive models. We fit two independent logistic regressions which take the demands as input variables and predict either success or correct choice. We also design a structural equation model using the `lavaan` library in R (Rosseel, 2012), with an identical structure to our measurement layout, serving as an alternative, *frequentist*, implementation to PyMC. Structural equation models (SEMs) in `lavaan` are significantly less flexible than PyMC models, offering less control over link functions and compensatory relationships between latent capabilities. Being frequentist, they do not provide posterior probability distributions or probabilities interpretable on a trial-by-trial basis, but they fit more quickly. Figure 8 shows that logistic regressions are generally more predictive than both our PyMC models and SEMs, but at the cost of explainability and being able to predict multiple response variables and their interaction under one model. SEMs and our measurement layouts behave similarly, with our PyMC models being more predictive on 9 out of 16 prediction tasks.

Figure 9 presents the estimated posteriors for three capabilities: object permanence, navigation, and visual acuity. All agents except children have low object permanence capability. In contrast, the children are poor navigators, compared to the reinforcement learning agents and particularly the heuristic agent. The heuristic agent is useful sanity check—it was designed to navigate only to observable goals. We were successfully able to recover the fact that it fails when goals are occluded but is nevertheless a good navigator. On visual acuity, it seems that all the reinforcement learning agents were heavily downsampling visual inputs, thus affecting their visual acuity compared to the random and heuristic agents. A measurement layout with the visual acuity capability ablated is also shown for children, since they were presented with tasks with no variance in goal size.

## 6  CONSTRUCTING EFFECTIVE MEASUREMENT LAYOUTS

For scalability and usability, it is prudent to have a generalised methodology for constructing measurement layouts. The core of our approach is to identify the *demands* of a task, and to compare these against the *capabilities* of the system being evaluated, while accounting for any sources of noise (*robustness*). For demands we use directly observable features of the task instance to ensure that demands are both easily measurable and objective properties of the task. We limit demands to meta-features that 1) should meaningfully affect performance (actual difficulty indicators) and 2) suspected to affect performance but should not (potential bias indicators). Each demand is related to a capability to provide a margin and typically will use a linking function of $\sigma(capability - demand)$. This approach is beneficial because it ties capabilities to bounded, probabilistic performance based on the demand. This also causes the capabilities to have the same unit as the demand, and allows us to interpret a capability with value $x$ as consistently succeeding on the demand with meta-feature $x$ in $50\%$ of instances, higher if capability exceeds demand, and lower otherwise. Of course, with more knowledge about the task, more intricate linking functions can be used to capture this knowledge succinctly.

In order to link demands and capabilities with task success, while accounting for behavioural robustness, we need to understand the dependency structure of the task; that is, which demands need to be met in order to succeed. In some cases, capabilities can be compensatory–meeting one demand

adequately can account for failing another demand, i.e., the system's capabilities compensate for each other. In other cases, they can be non-compensatory–both capabilities are necessary to reach some level of performance. In our experiments, our capabilities were non-compensatory, so we often opted for taking the product of probabilities. However, more complex relations such as taking the minimum or maximum of a set of probabilities is available by utilising a generalised mean (preserving differentiability of the link functions). The other major property that needs to identified in order to construct the measurement layout is the sources of possible bias. In principle, this is similar to creating the structure for capabilities: The meta-features that may affect performance need being identified, however, now those that are not necessary for the task are also required. A bias "performance" is now also included for each instance (how present was the bias in this instance, and in which direction was it directed?), but this is now included in the sigmoid function for the relevant capability it may affect: $\sigma(capability - demand + bias)$. This formulation is typical within IRT, for instance, the Rasch model (Rasch, 1960). Measurement layouts build on this formalisation by chaining them together within HBNs. Once all of the demands and biases are accounted for, a final combination can give an output for overall task performance.

Breaking a complex task down into demands, possible biases, and sources of noise is not always straightforward. Care, and domain knowledge of what constitutes success in the task, needs to be used to identify the core broad capabilities required, and how these break down to identifiable meta-features. We recommend an iterative approach: if a particular measurement layout cannot explain system behaviour reliably (with a low standard deviation on the resulting posterior distributions), then there is either too few evaluation instances, or the measurement layout is missing an important capability or bias.

## 7 RELATED WORK

**Psychometrics and latent variable modelling.** Measurement layouts are inspired by psychometric traditions that infer latent constructs from patterns of performance. Classic techniques such as Item Response Theory (IRT; Gustafsson & Undheim 1996) and Structural Equation Modelling (SEM; Keith & Reynolds 2010) infer latent abilities by analysing populations tested on relatively few items. These approaches typically assume homogeneous item types, linear link functions, and population-level data. In contrast, machine learning evaluation usually involves thousands of heterogeneous task instances but only a single subject (the model) of interest. Our method adapts psychometric ideas to this setting: we derive capability priors from domain knowledge, and infer a cognitive profile for an individual system by triangulating across varied task demands. This allows richer inference than task taxonomies or category aggregates, and avoids post-hoc interpretation of statistical factors as "capabilities." Related Bayesian frameworks such as HiBayES (Luettgau et al., 2025) focus on hierarchical GLM modelling of aggregate performance and uncertainty quantification; our emphasis is instead on *capability-oriented* inference from instance-level meta-features. The ADeLe framework (Zhou et al., 2025) also identifies capabilities in relation to task demands, but does not allow for the complex combination of demands with non-linear linking functions, limiting its explanatory power compared to measurement layouts.

**Cognitive modelling.** Hierarchical Bayesian models are widely used in cognitive science to explain human behaviour (Lee, 2011). These models are often simple at the individual level, designed to capture limited behavioural paradigms. By contrast, measurement layouts scale hierarchical Bayesian inference to complex, multi-demand tasks with thousands of instances, and support non-linear, non-compensatory interactions between capabilities. Our contribution is to treat capabilities as explicit latent variables linked to meta-features of tasks, rather than black-box latent factors.

**Evaluation frameworks in machine learning.** Recent ML evaluation efforts emphasise broad benchmarking rather than structured inference. Frameworks such as HELM (Liang et al., 2022), Dynabench (Kiela et al., 2021), and related initiatives highlight the need for dynamic and holistic evaluation, but typically report aggregated performance metrics or leaderboards. These aggregates are useful but opaque: they predict average success rates without explaining *why* a model succeeds or fails. Our approach is complementary: measurement layouts retain the generality of diverse benchmarks while providing explanatory, capability-level inferences that can predict behaviour on new tasks. This goes beyond static aggregate statistics, enabling interpretable and predictive evaluation.

## 8 DISCUSSION

Increasingly, machine learning systems are (pre-)trained once and deployed for a range of tasks. Task-oriented evaluation based on aggregated performance cannot robustly evaluate these systems across the many situations in which they might be deployed, nor predict behaviour for new tasks and changing distributions. However, we can often identify abstract capabilities that are not directly measurable, such as 'object permanence' in agents or 'handling negation' in language models. In the behavioural sciences this is a common approach, but many techniques assume a population of participants tested on a small number of tasks. Inferring capabilities from the behaviour of single individuals by contrasting their performance against the task demands is common in animal cognition research, but it is unusual to see complex inferences from thousands of items, as is common and required in machine learning and AI research. In this paper we have integrated several elements to make this inferential exercise possible and shown it can identify the capabilities and biases of each model—its cognitive profile—independently.

This unleashes a new way of evaluating general-purpose systems, by identifying areas of weakness or reasons for task failure. This is especially useful for hierarchical capabilities with multiple levels of dependency. We can speed up the debugging process by isolating skill-gaps or biases responsible for failure, and improve confidence in predictions of performance on new tasks. Instead of hoping that the average performance will hold on new tasks, our approach enables the use of task demands and the inferred cognitive profile to determine whether system deployment is safe and worthwhile. There are several limitations of this work. If many parameters must be inferred simultaneously then many evaluation instances are needed. We advocate incremental approaches, where each capability is inferred with subsets of tasks before moving to tasks with more complex dependencies.

Here we applied our method to the domain of agentic systems, but our method applies equally to any evaluation setting, including in natural language processing, computer vision, speech modelling, and reinforcement learning. Ultimately, the biggest challenge for the construction of measurement layouts is the need for detailed domain knowledge and robust evaluation batteries. Both can be costly, but are urgently needed to improve AI evaluation. Currently, evaluation is focused on advancing state-of-the-art performance on an increasingly small selection of benchmarks that often lack construct validity and are limited in scope (Koch et al., 2021; Raji et al., 2021) or do not actually measure what the benchmark designers originally intended (Browning & LeCun, 2023). Further care is needed when evaluating cognitive capabilities; it can be inappropriate to simply apply the same tests that would be used for human subjects (Mitchell, 2023) and there are numerous other pitfalls to avoid and good practices to adhere to (Ivanova, 2023).

A possible limitation of measurement layouts is the need for the domain expertise required to build encode the capability-demand relationship into a Bayesian structure. This can be time/expertise intensive. However, rather than being a purely negative factor, we believe this is part of a nuanced trade-off, enabling some key positives: First, it keeps the evaluation model interpretable and auditable: every node and edge in a layout has a clear meaning (e.g., "time under occlusion loads on object permanence"), rather than being the result of an opaque optimisation. Second, it lets us directly encode domain expertise and theory-driven constructs about capabilities and task demands, in the same way that psychometrics and cognitive science batteries are designed around hypothesised abilities such as working memory or object permanence. In many of our experimental settings (e.g., comparing RL agents to children on object permanence tasks), the choice of which capabilities to measure and how task features relate to them is intrinsically normative and theoretical, and we do not think this should be delegated entirely to an automatic procedure.

Our approach also highlights the need for a clear distinction between training datasets, which can be massive, unannotated, and built in an adversarial way (Kiela et al., 2021; Jia & Liang, 2017; Zellers et al., 2019), and evaluation batteries, which should be devised carefully perspective, and with systematic variation across the elements that might affect performance. Our vision is that, from well-constructed batteries such as those used here, building measurement layouts and triangulating predictive and explanatory capability profiles with them should become routine.

## REPRODUCIBILITY STATEMENT

The implementation of this work, including all measurement layouts, will be made available upon publication with a GitHub repository. This will include, for each of the experiments, Jupyter notebooks that can run our experiments with minimal effort. Any data files will also be provided.

## USE OF LARGE LANGUAGE MODELS

Large language models were used to assist with grammar and sentence formulation in select places. They were also used to identify additional research literature and write python code for plotting some figures.

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

## A    PSYCHOMETRIC UNDERPINNINGS

Measurement layouts are informed by, but distinct from, classic psychometric approaches:

1. We do not simply break out performance by 'categories'. Related work employs a 'taxonomy of tasks' (see e.g., Table 1 in Liang et al., 2022), each placed into one or more categories. This limits the inferential capacity and predictability of these category aggregates for new tasks. Instead, **we evaluate based on instance-level demands**.

2. Unlike psychometric approaches such as factor analysis (FA), structural equation modelling (SEM) or item response theory (IRT), we do not rely on populational data (Gustafsson & Undheim, 1996). Instead, **we infer the cognitive profile of a single subject from their performance data alone**.

3. We do not simply extract latent variables statistically and then post-hoc interpret them as capabilities, which has been done hierarchically elsewhere (e.g., the Cattell-Horn-Carroll model (Keith & Reynolds, 2010)). Instead, we first identify prerequisite capabilities based on domain knowledge, and following this, derive general linking functions that allow us to infer and measure the subject's latent capabilities from the demands of the task.

4. Probabilistic models currently used in cognitive modelling (Lee, 2011) are often simple at the level of the individual and cannot convey the required probabilistic expressions connecting individual performance, capabilities and task features. Instead, **complex interactions allow for Bayesian triangulation which we use for informative evaluation**.

Together, these differences adapt psychometric reasoning to the machine learning setting, where evaluation involves many heterogeneous task instances but often only a single subject of interest.

## B EXPERIMENTAL MATERIALS

To illustrate how a measurement layout identifies cognitive profiles, we first select a domain with simple task characterisations that can be straightforwardly related to cognitive profiles to predict performance. We chose the Animal-AI (AAI) (Beyret et al., 2019; Voudouris et al., 2025) environment as this domain for the scalability of task complexity it provides.

Animal-AI is an environment for training and testing AI systems. It is built in the Unity engine (Juliani et al., 2018). Animal-AI provides a single agent with a first-person perspective in a 3D environment. The environment can contain a number of pre-specified objects, representing rewards, obstacles, tools, or dangers. Tasks can be defined using a domain-specific language to procedurally generate a large number of task instances. The agent's interaction with the world is limited to moving around in 3D space, though it can exploit physical interactions (such as pushing certain objects) to generate more complex behavioural sequences.

In the main text, we explore two evaluation test batteries developed in AAI. In Experiment 1 (Section 3), we design some simple tasks for measuring the navigation and visual acuity capabilities of hand-designed agents. In Experiments 2 and 3 (Sections 4 and 5), we use the *Object-Permanence in Animal-Ai: GEneralisable Test Suites* (O-PIAAGETS) test bed (Voudouris et al., 2022). We describe each test battery in turn.

### B.1 SIMPLE NAVIGATION AND VISUAL ACUITY TEST BATTERY

To demonstrate the utility of measurement layouts, we use a simulation data from agents with known capabilities. The aim of the measurement layout is to infer an agent's visual acuity (its capability to see) and its navigation ability (its capability to navigate towards rewarding objects).

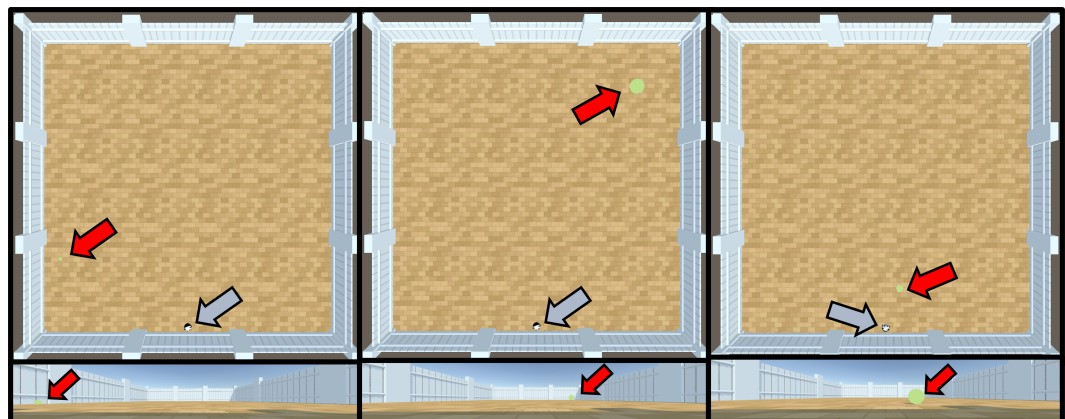

Figure 10: A series of tasks with the agent in a fixed position and goals of different sizes in 1 of 100 different positions in the arena.

In the Animal-AI Environment, a battery of tasks can be easily defined. The agent is placed in the arena at a fixed position and random starting rotation, along with a single rewarding *GoodGoal* of a certain size placed at some location in the arena. The agent must locate the goal and navigate towards it. The agent passes the instance if they obtain the goal, and fails otherwise. We procedurally generated 1000 instances of arenas in Animal-AI, with the agent in a fixed position at $(20, 0, 0.5)$ and goals in 100 evenly spaced positions across the arena. There are 10 possible $x$-coordinates and 10 possible $z$-coordinates, at $[2, 6, 10, 14, 18, 22, 26, 30, 34, 38]$ for each. The Euclidean distance between the centre of the agent and the centre of the goal was computed, encoding the distance demand. There were 10 different sizes for each goal, $[0.2, 0.4, 0.6, 0.8, 1.0, 1.2, 1.4, 1.6, 1.8, 2.0]$. Figure 10 presents three tasks with goals of different sizes placed in different positions.

The smaller the goal or the further it is from the agent, the higher the visual acuity required for the agent to see it. This is because the 'retinal image' of the object decreases for smaller, more distant goals. There are two task demands, therefore, *goal distance* and *goal size*. We want the latent *visual acuity* capability estimate to be higher if the agent tends to pass instances where the goal is further

away and/or smaller. As such, we want the overall demands of the instance (some combination of size and distance) to be *higher*. *Retinal size*, which can be defined as size divided by distance, decreases for 'harder' tasks. So an alternative is to invert retinal size, by dividing distance by size. This value increases when the goal has a smaller retinal size from the starting point. In summary, by defining the visual acuity demand as the inverse of the retinal size, we can produce a scale for latent visual acuity estimates with interpretable units, and therefore, interpretable magnitudes. Figure 2 b) presents a measurement layout relating these two task demands to performance.

## B.2 O-PIAAGETS

To demonstrate the Measurement Layout framework's ability to handle more complex tasks and infer complex psychological capabilities, we also apply it to a subset of the O-PIAAGETS test battery (Voudouris et al., 2022). O-PIAAGETS is also built in the Animal-AI environment and is designed to assess object permanence. O-PIAAGETS consists of three broad task types from the animal and developmental sychological literature for assessing object permanence and a set of control tasks testing for basic skills that are required to solve more complex OP tasks.

In all experiments, two experimental paradigms were selected from O-PIAAGETS: the Primate Cognition Test Battery and Chiandetti & Vallortigara Tests of Intuitive Physics. The corresponding instances from the control battery for these experimental paradigms were also used, as well as a series of simple instances. We describe each of these tasks below. In all figures, red arrows denote the location of rewards and grey arrows denote the location of the agent.

### B.2.1 BASIC CONTROLS

A series of instances are used that involve the basic objects in Animal-AI 3 in simple combinations. In this chapter, these are collectively called the *Basic Control Tasks*. This includes instances where the goal rolls from one side to the other, or is on a blue platform obtainable by navigating up a ramp. There are some tasks that require detouring around opaque and transparent walls and death zones, or to make simple choices (see Figure 11).

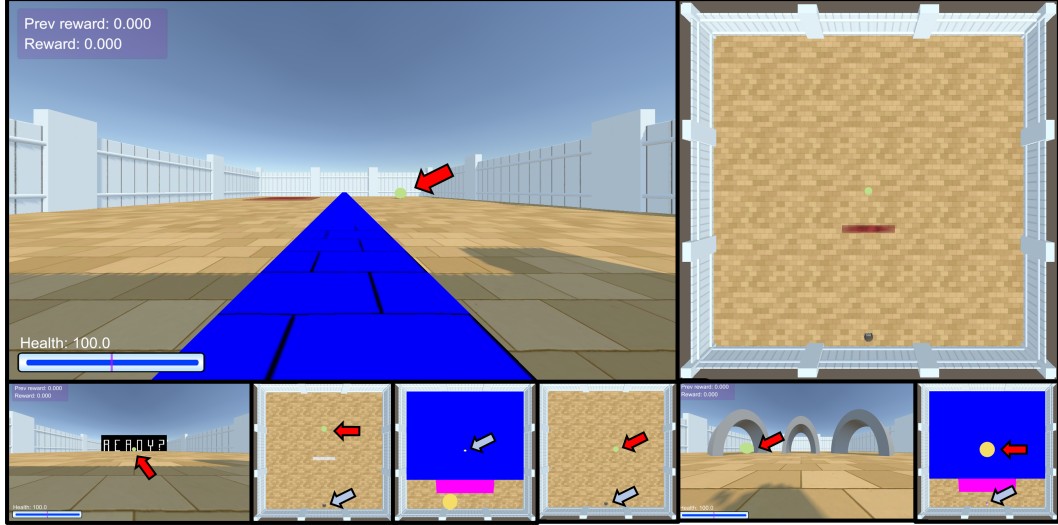

Figure 11: Basic Control Tasks. Top Left: A forced choice task—the agent is spawned on a platform with a green reward on the right and a death zone on the left. Top Right: An avoidance task—the agent must navigate around the death zone to obtain the green reward. Bottom: A selection of further simple tasks.

### B.2.2 CHIANDETTI & VALLORTIGARA TESTS OF INTUITIVE PHYSICS

The instances in this paradigm are referred to collectively in this chapter as the *CV Chick Tasks*. Figure 12 presents one task. Here, the agent is frozen on a platform, watching a reward roll directly

away from it through a death zone. Then the lights go out, during which time the goal is deflected behind the wall on the left. When the lights come back on the agent is unfrozen and must infer that the goal is occluded on the left. This demands OP, because the agent must behave as though the object continues to exist even though it is (a) not visible, and (b) not conceivably occluded where it was last observed (in the death zone). This task varies: the colour of the occluding wall, the side on which the goal is occluded, and the presence of another wall on the other side. These walls are either: transparent, thinner than the width of the goal, or lower than the height of the goal, meaning that they would not be able to occlude the goal were it there. There are two special cases of this test. In one, there are two identical occluders on each side and no lights out period. The agent watches the reward roll behind one occluder, and must navigate towards it. In the second, there are two identical occluders which fall backwards during the lights out period. One occluder falls (almost) flat to the ground, meaning that the goal could not be behind it or under it. The other occluder falls only slightly, meaning the goal could be (and is) underneath it. These tasks also vary in wall colour and whether the goal is to be found on the right or the left.

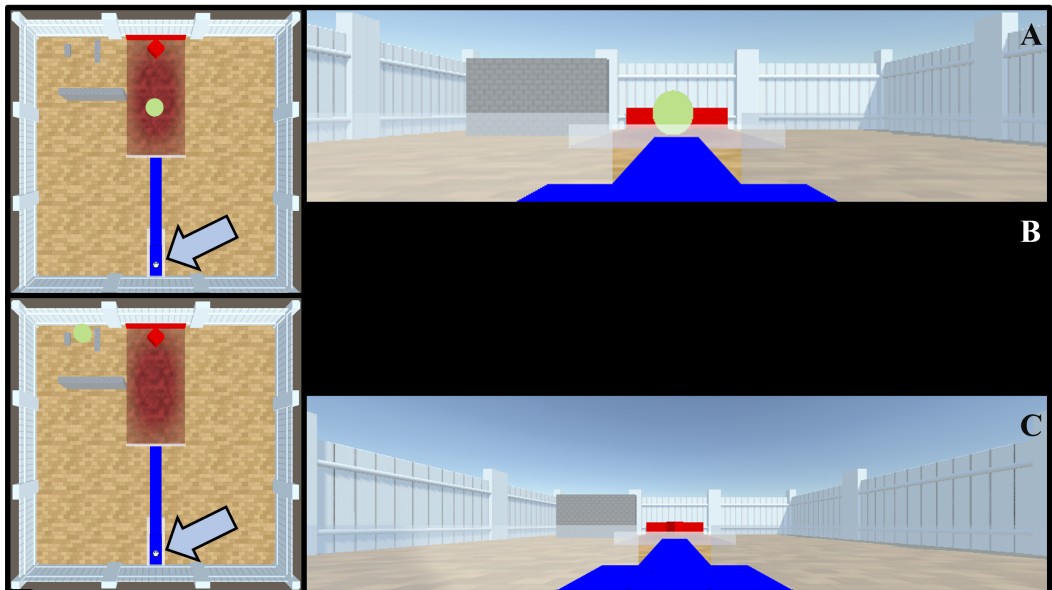

Figure 12: An instance of a CV Chick Task from the Object Permanence test battery. A–C: Three stages of the task.

Figure 13 presents a version of a CV Chick Task that does not demand OP. The arena is constructed identically, with the agent on a platform and a death zone in the middle. The agent is frozen, and watches the goal roll to the left in front of a wall. There are variants with and without lights out periods. The goal is therefore never occluded. These tasks vary: the size of the goal, the colour of any wall objects, the side that the goal rolls to, the presence of a lights out period, the height or width of potential occluders, and the presence of transparent walls. There are variants of the tasks with falling walls, except that the wall falls forwards and the goal is always visible.

### B.2.3 THE PRIMATE COGNITION TEST BATTERY

The two kinds of task inspired by the Primate Cognition Test Battery (PCTB) from O-PIAAGETS were included in this study. Instances from the first kind are collectively called *Three-Cup Tasks* here. The left image of Figure 14 presents an instance for testing OP. The agent is spawned in front of three ramps, separated by coloured walls. These simulate the cups under which the experimenter hides rewards in the original Primate Cognition Test Battery tasks. Goals fall from a height into one or more of the cups, dropping behind a wall so that they become occluded. In cases where only one goal is dropped, the ramps allow the agent to enter the cup but not exit it. When two goals drop, there are two ramps in cups with multiple goals, but only one ramp for the incorrect ramp, so if the agent makes the incorrect choice, they fail. These tasks vary: the distance of the agent from the

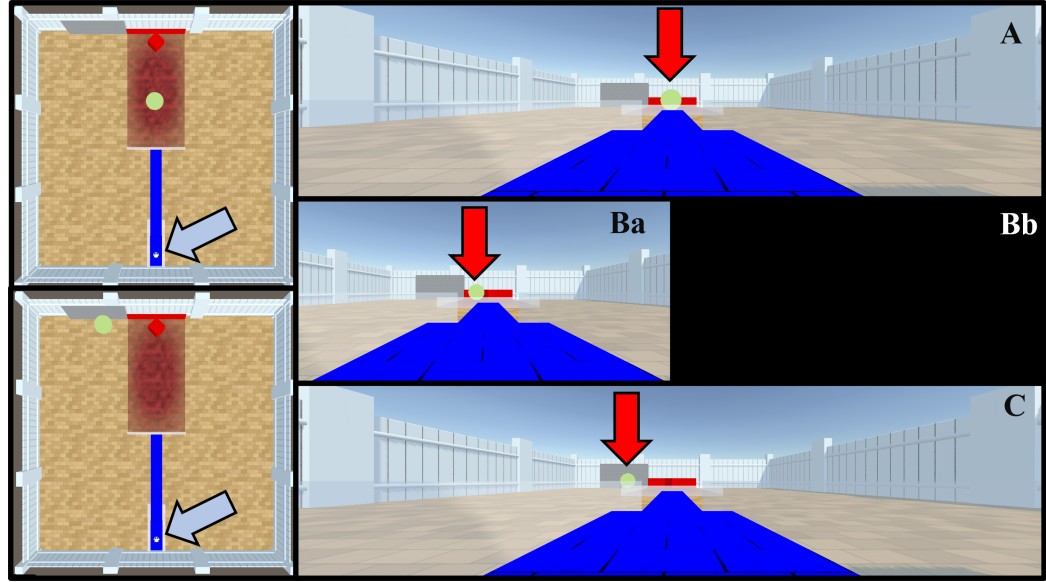

Figure 13: An instance of a CV Chick Task from the Room Practice Control test battery, which does not test for OP. A–C: Three stages of the instance. Every instance has a lights out and a no lights out variation.

cups, the type of goal (yellow or green), the size of the goal, the colour of the occluding walls, the number of goals, and the presence of death zones next to the ramps (to make navigation harder).

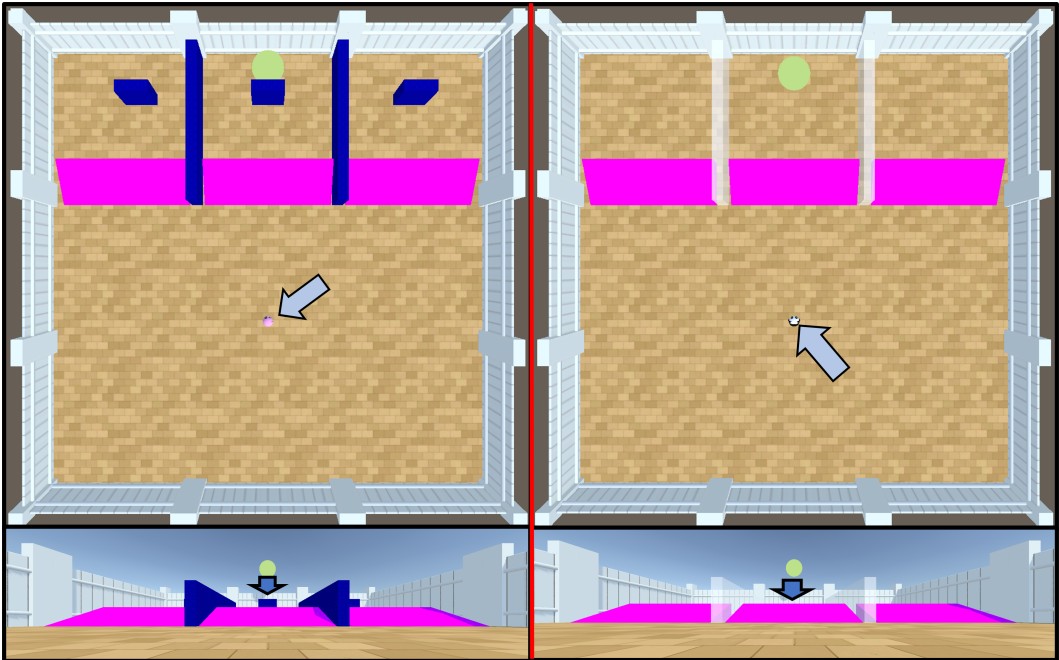

Figure 14: Instances of a PCTB Cup Task. Left: A version of the task from the Object Permanence test battery. Right: A version of the task from the Room Practice Control test battery.

The right image of Figure 14 presents a control version of the *PCTB Three-Cup Task*, which does not require OP. In this case, the walls between the cups are transparent, and there are no walls behind which the goal is occluded once it has dropped. These tasks vary: the distance of the agent from the cups, the type of goal (yellow or green), the size of the goal, the colour of any walls in the arena,

the number of goals, the presence of death zones next to the ramps, the presence of ramps, and the presence of transparent walls in front of the goal(s).

Instances from the second kind of PCTB task are collectively called *PCTB Grid Tasks* here. The agent is spawned atop a large ramp, with a view over a series of holes in the floor, demarcated by white rims. A single green goal drops down one of the holes, and the agent must navigate to the correct one. In the left image of Figure 15, the goal drops down one of 12 holes. Once it has dropped, it is no longer visible, testing OP. In the right image of Figure 15 is a version of the Grid Task that does not test OP. Here, the holes are shallow, goals continue to drop from a height, but remain visible in their position. Both the OP and control tasks vary: the number of holes (4, 8, 12), the size of the goal, and the hole into which the goal drops.

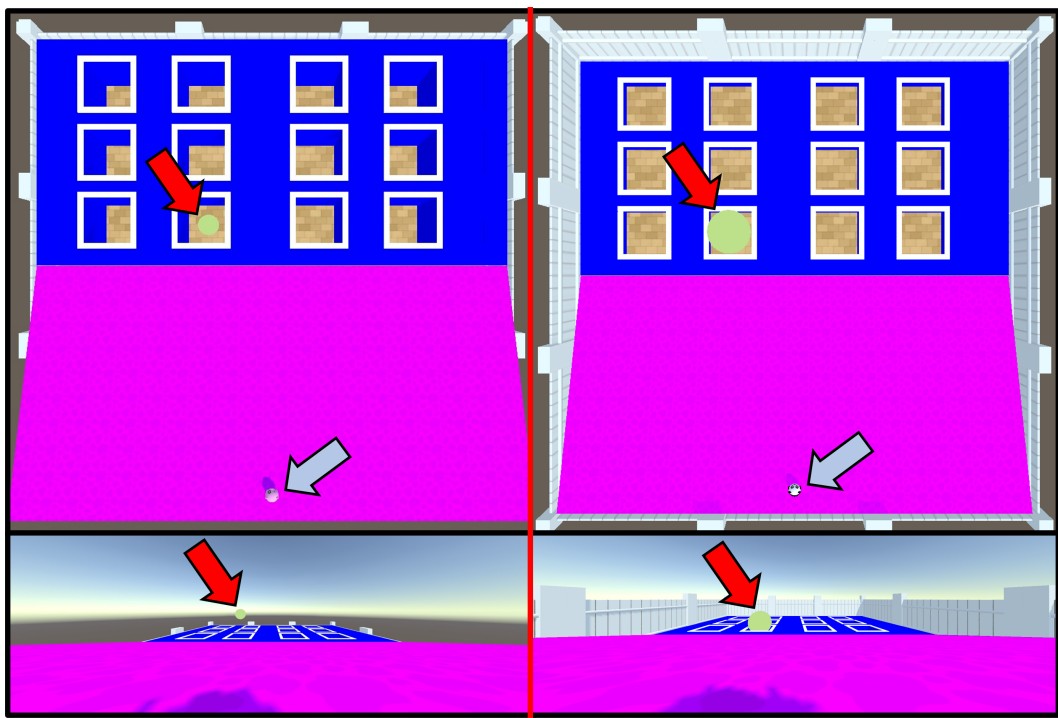

Figure 15: Instances of a PCTB Grid Task. Left: A version of the task from the Object Permanence test battery. Right: A version of the task from the Room Practice Control test battery.

### B.2.4 TASK OVERVIEW

These tests are relatively simple in terms of what is required of the agent compared to some other tasks in O-PIAAGETS. They also contain a large number of variants, controlling for several alternative hypotheses. Moreover, the PCTB Tasks and the CV Chick Tasks are an important dyad, because they control for distinct hypotheses about behaviour. One sophisticated policy that could lead to good performance on PCTB Tasks would be to navigate to where the goal was last observed. This is not necessarily the same as having OP: the agent need not have any representation that the *same* goal exists at the last observation location; they could represent the goal as having disappeared while having learnt that they tend to get rewarded with a 'new' goal when they navigate to the last observation location. This would be a successful strategy for the PCTB Tasks, but not the CV Chick Tasks, which require that the agent represent the goal as continuing to exist, so that they can make the inference that it must be occluded behind one of the walls.

In the other direction, the CV Chick Task could be solved by a policy that navigates behind the largest wall in the scene. This policy would lead to success on many CV Chick tasks, but it would not lead to success on the PCTB tasks, where there are either three walls of equal width in the case of the Three-Cup tasks, or no walls at all in the case of the Grid tasks. Thus, using both the PCTB tests and the CV Chick tests is a particularly useful subsection of O-PIAAGETS to focus on.

## C  EVALUATION SUBJECTS

For each experiment, we used a different set of subjects (agents) for evaluation. We describe each in turn.

### C.1  EXPERIMENT 1: A SIMPLE NAVIGATION TASK

Each agent has two parameters: $p$, where $p \times p \times 3$ is the size of the pixel input from the environment with three colour channels (R, G, B), and $\nu$, where $1 - \nu$ is the probability of taking an action at a time step and $\nu$ is the probability of taking no action (i.e., staying still). The agent rotates left in its starting position. If there are green pixels to the right or left of the centre of the image, it rotates accordingly until they are in the centre, with probability $1 - \nu$. If green pixels are present in the centre of the image, it navigates forward with probability $1 - \nu$. The pixel input size constrains the agents visual acuity, by limiting what it can see in the environment. The navigation noise constrains the ability of the agent to navigate towards a reward, with higher values lowering the agents navigational ability. We simulate 100 agents on our 1000 tasks, using pixel inputs ranging from $4 \times 4$ to $40 \times 40$ in increments of 4 and navigational noise ranging from 0 to 0.9 in increments of 0.1.

### C.2  EXPERIMENT 2: AN OBJECT PERMANENCE TASK

Object Permanence (OP) is the understanding that objects continue to exist when they go out of view. Behaviourally, an agent has object permanence if they behave *as though* objects continue to exist even when they cannot directly perceive them. It is a key component of physical common-sense (Shanahan et al., 2020; Lake et al., 2017; Baillargeon et al., 2010a), allowing biological agents to more successfully predict and interact with their environment. Object permanence is well studied in cognitive science, where several experimental designs are used to detect its presence in human and non-human animals (Scholl, 2007; Flombaum & Scholl, 2006; Hare & Tomasello, 2005; Chiandetti & Vallortigara, 2011). Many of these paradigms are represented in O-PIAAGETS, which contains over 22,000 instances that test OP (as well as over 250,000 control instances that test other skills necessary to complete the main battery of instances). We selected three paradigms that test for *allocentric object permanence*, where objects are occluded independently of the observer's actions. We also include basic control tasks within the test-suite to evaluate the agent's non-OP capabilities and make the inference process more robust. Details about the test-bed design criteria, the included tasks, and why object permanence is difficult to identify are given in Appendix E.

This scenario tests whether our methods can recover complex cognitive profiles from performance data of agents for which we know the ground truth. To do so, the the authors of this paper were divided into two teams, *Team A* and *Team B*. Team A generated the synthetic dataset, while Team B defined the measurement layout and attempted to recover the cognitive profiles of the synthetic agents.

In total, we included 2188 unique instances from O-PIAAGETS, varying along the 12 dimensions outlined in Table 1. Note the asterisks in the table, denoting that a particular feature was not used in the measurement layouts used for evaluation in this paper. This occurred because when constructing the measurement layout, Team B either did not realise the feature was present in the environment or because it was missed as a possible source of bias or capability. Such discrepancies are an unsurprising result of the two-team blind set-up, but also mirrors a situation in which the full ground truth of a population is unnkown. Nevertheless, this actually highlights a strength of our evaluation framework; imperfect measurement layouts can still be useful for both inferring agent capabilities and predicting future agent performance.

Team A generated 30 synthetic agents of four different types to ensure a diversity of behaviour. *Reference Agents* act as reference points, e.g., a perfect agent, an agent with $x\%$ success for various values of $x$. *Achilles Heel Agents* have low ability on some dimension, but otherwise perform well and can be said to have object permanence. *Context-Specific Agents* have mixed OP performance, doing well in some specific tests/contexts but not others. *Fraudster Agents* behave in a manner that might look like object permanence, but are actually using rigid heuristics such as "go to last seen location of reward, then search" (Užgiris & Hunt, 1975). Tables 5, 6, and 7 in Appendix E.1 show the specifics of each agent. This information was not disclosed to Team B.

| Dimension | Description | Values |
|---|---|---|
| Reward Visibility* | Is the reward visible at the choice point? | 0, 1 |
| Time Under Occlusion | Time reward is occluded | $[0, 4]$ |
| Positions | Number of alternative reward places | $[1, 11]$ |
| Reward Distance | Manhattan distance to the goal | $[0, 56]$ |
| Right/left Position | Is reward left, right or straight from the agent? | $\{-1, 0, 1\}$ |
| Reward Size | How large is the reward? | $[0.5, 4]$ |
| Occluder Presence | Are there any occluders hiding the rewards? | $\{0, 1\}$ |
| Lava Presence | Is there any lava in the arena? | 0, 1 |
| Platform Presence | Are there any platforms in the arena? | 0, 1 |
| Ramp Presence | Are there any ramps in the arena? | 0, 1 |
| Transparent Walls | Are there any transparent walls in the arena? | 0, 1 |
| Occluder Redness* | What is the R value of occluders in the arena? | $[0, 255]$ |
| Occluder Greenness* | What is the G value of occluders in the arena? | $[0, 255]$ |
| Occluder Blueness* | What is the B value of occluders in the arena? | $[0, 255]$ |

Table 1: Object Permanence Task Characterisation details: The dimensions on which the instances vary. Dimensions with an asterisk are not included in the measurement layout presented in this paper.

### C.3 EXPERIMENT 3: EXTENDING MEASUREMENT LAYOUTS TO REAL DATA

We use the dataset from Voudouris et al. (2024) to fit measurement layouts for realistic data collected from deep reinforcement learning agents and human children, as well as a heuristic agent and a random action agent.

We use the random agent to define a baseline for chance performance. This agent takes one of the nine available actions available in the Animal-AI Environment and repeats it for a number of steps, $S$ sampled from a uniform distribution: $S \sim U(0, 20)$, after which it randomly selects a new action. It executes an action for that many steps before selecting a new one. There are no biases to select certain actions.

We use the heuristic agent as a more meaningful baseline, as an agent that lacks object permanence, but possesses many other capabilities that would lead to success in the Animal-AI Environment. We do not expect random walkers to fail tests of OP because they lack only the capability of OP. Rather, we expect them to fail because they lack many other capabilities, such as the capability to move towards an appetitive reward or away from an aversive stimulus (goal-directedness). The heuristic agent follows a small set of rules and heuristics. This agent can view objects in the $x$-$z$ plane that are in the forward-facing $60°$ of its view using raycasts. When it detects a green or a yellow goal in this viewing angle, it orients itself towards it and moves forward to obtain it. When it detects a red goal or a death zone, it orients itself away from it and moves backwards. If the agent is stationary or if there is a wall in front of it, the agent moves either forwards and left or forwards and right with a probability $p$=0.5, perseverating that movement subsequently to navigate around the object until the agent is no longer stationary or there is no longer a wall in front of it. If the agent does not detect any goals, death zones, or walls, and if it is not stationary, it perseverates its previous action.

Data from children aged 4–7 years old performing on a subset of the O-PIAAGETS tasks was also used as a meaningful baseline for an agent that is error-prone but possesses object permanence and other ancillary capabilities necessary to complete tasks in Animal-AI Baillargeon et al. (2010b); Užgiris & Hunt (1975); Piaget (1923). Data from children was collected between January and March (inclusive) 2023 by the authors of Voudouris et al. (2024), with a minimum target sample size of 30. The study was conducted at a university, in the presence of researchers and the participants' guardians. Guardians were provided with an information sheet and the opportunity to consent to their child's participation in the study. If they consented, the study was initiated. Participants watched a three-minute video introducing the study as a game called 'Get the Fruit!'. Participants were invited to assist 'Farmer John' in finding all the green and yellow 'apples' in the farmyard,

while avoiding any poisonous red fruit and 'large puddles of lava rain.' They were informed that they would receive stickers in return for fruit collection, although they actually received one sticker every ten trials, regardless of whether they successfully found the goal, and an extra sticker upon completion of the study. After the video, participants commenced play with the Animal-AI Environment, interacting with it through a small hand-held controller connected to a computer. First, they played a 'tutorial' round, consisting of 11 tasks. They could play these tasks as many times as they wished, and ask for help throughout. Then, they played the 'test' round, consisting of 38 Test tasks. These tasks were ordered randomly, with half of participants playing the fixed random order and the other half playing the reverse of that order. Every 10 tasks, the participant could take a break from the game and receive their stickers. The break consisted of a simple task with a single large green goal and no time limit. When the participant was ready to continue playing, they simply had to navigate to the goal to initiate the next batch of tasks. During the test round, guardians filled out a short survey about the participant, with questions on age, gender, and videogame playing habits. Upon completion of the test round, participants and their guardians were debriefed and offered a book voucher for their participation.

The deep reinforcement learning agents, Dreamer-v3 and PPO, were each trained on 5 different curricula, resulting in 5 different trained agents (see Table 2).

Table 2: The 5 different trained agents for each architecture (Dreamer-v3 & PPO; 10 total), their training curricula, and the number of training steps. Stratification was done by experimental paradigm.

| Agent Name | Training Curriculum | Training Steps (Millions) |
|---|---|---|
| PPO/Dreamer 1 | All Basic tasks | 2M |
| PPO/Dreamer 2 | All Basic tasks + a stratified random sample of 300 Control tasks | 4M |
| PPO/Dreamer 3 | All Basic tasks + all Control tasks (in 3 randomly sampled batches) | 8M |
| PPO/Dreamer 4 | All Basic tasks + a stratified random sample of 300 Control tasks + a stratified random sample of 300 Test tasks | 6M |
| PPO/Dreamer 5 | All Basic tasks + all Control tasks (in 3 randomly sampled batches) + a stratified random sample of 300 Test tasks | 10M |

## D  Measurement Layout Specifications

### D.1  Experiment 1: A Simple Navigation Task

The measurement layout for the simple navigation task consists of two capability nodes (navigation, visual acuity) and two meta-feature nodes (goal distance, goal size), as depicted in Figure 2B. Each of the capability nodes has an associated instance-performance node, an intermediate non-observable node (INON). To compute probabilities at these nodes, we use the sigmoid function over the difference between estimated capability and the current meta-feature value. For visual acuity, this is the goal distance divided by the goal size. We apply the natural logarithm to this difference to improve numerical stability. For navigation, the meta-feature is simply the goal distance. We assume a non-compensatory relationship between navigation and visual acuity, and therefore multiply these marginal INON probabilities to condition the Bernoulli distribution over binary (success/failure) outputs. We use the half-normal prior over capabilities with a $\sigma$ equal to half of the maximum relevant meta-feature or meta-feature combination.

### D.2  Experiment 2: An Object Permanence Task

The measurement layout for OP consists of seven capability nodes, one bias node, and one robustness node (stochastic noise on success/failure). Again, each of the capability and bias nodes has

| Name | Type | Range/Values | Prior/Linking Function |
|---|---|---|---|
| rampPresence | Meta-feature | $\{0,1\}$ | |
| lavaPresence | Meta-feature | $\{0,1\}$ | |
| platformPresence | Meta-feature | $\{0,1\}$ | |
| goalDistance | Meta-feature | $[0,56]$ | |
| goalSize | Meta-feature | $[0.4,5]$ | |
| rightleftPosition | Meta-feature | $\{-1,0,1\}$ | |
| occluderPresence | Meta-feature | $\{0,1\}$ | |
| timeUnderOcc | Meta-feature | $[0,4]$ | |
| numberOfPositions | Meta-feature | $[1,11]$ | |
| OPAbility | Capability | $[0,48.4]$ | $Uniform(0, memoryAbility \times maxPositions)$ |
| memoryAbility | Capability | $[0,4.4]$ | $Uniform(0,4.4)$ |
| visualAbility | Capability | $[0,6]$ | $Uniform(0,6)$ |
| rampAbility | Capability | $[0,1]$ | $Uniform(0,1)$ |
| lavaAbility | Capability | $[0,1]$ | $Uniform(0,1)$ |
| platformAbility | Capability | $[0,1]$ | $Uniform(0,1)$ |
| flatNavAbility | Capability | $[0,56]$ | $\mathcal{N}(0,1)$ |
| rightleftBias | Bias | $[-\infty,\infty]$ | $\mathcal{N}(0,1)$ |
| noiseLevel | Robustness | $[0,1]$ | $Uniform(0,1)$ |
| rampPerformance | INON | $[0,1]$ | $\mu(rampAbility, rampPresence)$ |
| platformPerformance | INON | $[0,1]$ | $\mu(platformAbility, platformPresence)$ |
| lavaPerformance | INON | $[0,1]$ | $\mu(lavaAbility, lavaPresence)$ |
| flatnavPerformance | INON | $[0,1]$ | $\sigma(flatNavAbility - goalDistance + rightLeftEffect)$ |
| navigationPerformance | INON | $[0,1]$ | $rampPerformance \times platformPerformance \times lavaPerformance \times flatnavPerformance$ |
| memoryPerformance | INON | $[0,1]$ | $\sigma(memoryAbility - timeUnderOcc)$ |
| OPPerformance | INON | $[0,1]$ | $memoryPerformance \times \sigma(48.4 - ((48.4 - (OPAbility - 4 \times numPositions)) \times occluderPresence))$ |
| visualAcuityPerformance | INON | $[0,1]$ | $\sigma(visualAbility - (5 - goalSize))$ |
| rightleftEffect | INON | $[-\infty,\infty]$ | $rightLeftBias \times rightLeftPosition$ |
| taskPerformance | Observed | $[0,1]$ | $p \sim Bernoulli(\omega[noiseLevel](OPPerformance \times navigationPerformance \times visualAcuityPerformance, \nu))$ |

Table 3: Nodes in the Object Permanence Measurement Layout, including priors and linking function.

an associated instance-performance node, an intermediate non-observable node (INON). The noise is applied at the end to the final performance probability. A full description is given in Table 3 including all the value ranges, distributional priors and linking functions.

The function $\sigma$ is the standard logistic function while $\omega$ is a parameterised weighting function, where the value $\nu$ represents a noise prior that represents a constant model set to $1 -$ average performance. The function $\mu$ is a margin, defined as

$$\mu(a,b) = 1 - ((1-a) \cdot b)$$

where $a$ is an ability and $b$ a binary feature. This means that when the binary feature is 0 then the margin represents $p(success) = 1$. If the binary feature $= 1$ then $p(success) = a$. The units of some of the abilities derive from the meta-features, like the memory ability, which is seconds, as it comes from the opposing metafeature timeUnderOcc. Similarly, visualAbility has the same units as goalSize, and flatNavAbility has the same units as goalDistance. OPAbility comes from a product between timeUnderOcc and numPositions so its unit is seconds times positions. Clearly, the most complicated linking function is the one for OPPerformance, which basically is opposed to (subtracted by) the maximum time under occlusion (4) times the number of positions. This is subtracted from the maximum OPAbility (48.4) and multiplied by the occluderPresence which is subtracted again from the maximum OPAbility (48.4). A logistic function is applied as usual but then this is multiplied by memoryPerformance, representing the dependency between object permanence and memory.

### D.3 EXPERIMENT 3: EXTENDING MEASUREMENT LAYOUTS TO REAL DATA

To accommodate the data from Voudouris et al. (2024), we updated the measurement layout, which is presented in Figure 16.

There are three central capabilities: OP, navigation, and visual acuity. There are four further capabilities that are pertinent to navigation: the capability to avoid death zones, and the capabilities to navigate to goals placed to the left, directly ahead, and to the right of the agent's starting position. A full description is given in Table 4 including all the value ranges, distributional priors and linking functions.

## E EVALUATING THE PRESENCE OF OBJECT PERMANENCE

Object Permanence (OP), and similar cognitive capabilities such as episodic memory and theory of mind, are difficult to robustly identify in AI systems because there are often several competing

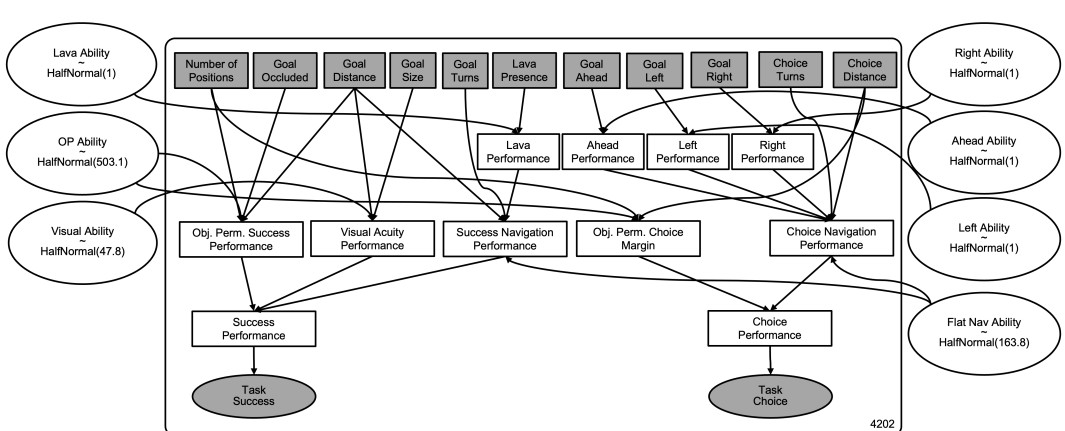

Figure 16: The updated measurement layout to accommodate the meta-features and observed variables collected in Voudouris et al. (2024).

| Name | Type | Range/Values | Prior/Linking Function |
|---|---|---|---|
| Number of Positions | Meta-feature | $\{0, 12\}$ | |
| Goal Occluded | Meta-feature | $\{0, 1\}$ | |
| Goal Distance | Meta-feature | $[9, 118]$ | |
| Number of Turns To Goal | Meta-feature | $[0, 13]$ | |
| Choice Distance | Meta-feature | $[1, 53.5]$ | |
| Number of Turns To Choice | Meta-feature | $[0, 6]$ | |
| Goal Size | Meta-feature | $[0.5, 5]$ | |
| Death Zone Presence | Meta-feature | $\{0, 1\}$ | |
| Goal Ahead | Meta-feature | $\{0, 1\}$ | |
| Goal Left | Meta-feature | $\{0, 1\}$ | |
| Goal Right | Meta-feature | $\{0, 1\}$ | |
| OPAbility | Capability | $[0, \infty)$ | $HalfNormal(503.1)$ |
| Flat Navigation Ability | Capability | $[0, 4.4]$ | $HalfNormal(163.80)$ |
| Visual Acuity | Capability | $[0, \infty)$ | $HalfNormal(47.80)$ |
| Lava Ability | Capability | $[0, \infty)$ | $HalfNormal(1)$ |
| Ahead Ability | Capability | $[0, \infty)$ | $HalfNormal(1)$ |
| Right Ability | Capability | $[0, \infty)$ | $HalfNormal(1)$ |
| Left Ability | Capability | $[0, \infty)$ | $HalfNormal(1)$ |
| OPSuccessPerformance | INON | $[0, 1]$ | $\sigma(OPAbility, GoalOccluded * (GoalDistance - ChoiceDistance) * NumberOfPositions)$ |
| OPChoicePerformance | INON | $[0, 1]$ | $\sigma(OPAbility, GoalOccluded * ChoiceDistance * NumberOfPositions)$ |
| FlatNavGoalPerformance | INON | $[0, 1]$ | $\sigma(FlatNavAbility, GoalDistance * NumberTurnsGoal)$ |
| FlatNavChoicePerformance | INON | $[0, 1]$ | $\sigma(FlatNavAbility, ChoiceDistance * NumberTurnsChoice)$ |
| visualAcuityPerformance | INON | $[0, 1]$ | $\sigma(visualAbility - \log(goalDistance/goalSize))$ |
| lavaPerformance | INON | $[0, 1]$ | $\mu(lavaAbility, lavaPresence)$ |
| AheadPerformance | INON | $[0, 1]$ | $\mu(AheadAbility, AheadPresence)$ |
| LeftPerformance | INON | $[0, 1]$ | $\mu(LeftAbility, LeftPresence)$ |
| RightPerformance | INON | $[0, 1]$ | $\mu(RightAbility, RightPresence)$ |
| SuccessPerformance | Observed | $[0, 1]$ | $p \sim Bernoulli(FlatNavGoalPerformance \times OPSuccessPerformance \times visualAcuityPerformance \times lavaPerformance)$ |
| ChoicePerformance | Observed | $[0, 1]$ | $p \sim Bernoulli(FlatNavChoicePerformance \times OPChoicePerformance \times visualAcuityPerformance \times AheadPerformance \times LeftPerformance \times RightPerformance)$ |

Table 4: Nodes in the Object Permanence Measurement Layout, including priors and linking function.

alternative explanations for why an agent is succeeding or failing at a particular task. We offer Bayesian triangulation as a solution to this problem.

Imagine a simple task where a reward is observed to roll leftwards behind an occluder, and go fully out of sight. An agent with OP ought to pass this task, since it: (a) wants to obtain the reward, (b) understands that the reward continues to exist despite not being able to see it, and (c) is able to infer the location of the occluded reward based on its previously observed trajectory. However, an agent with OP might fail this task, if for instance it is bad at navigating, and takes a very circuitous route to its goal and runs out of time before it can reach the reward. Conversely, an agent *lacking* OP might pass this task, simply because it has been rewarded previously for going forwards some number of steps and then turning left. This is the problem of *the underdetermination of theory by behaviour* (Stanford, 2006), in which an observed behaviour can be explained in multiple ways. The use of well-informed measurement layouts based on internally valid test batteries allows us to more robustly eliminate incorrect alternative explanations and triangulate on the underlying causal explanations of AI system behaviour.

There are many competing desiderata to consider when designing a test-suite for the evaluation of OP in AI systems.

1. Cognitive capabilities such as OP can be difficult to place on an ordinal scale. This is because it is often not clear what affects behavioural performance in humans and other animals, and to what extent certain instance dimensions are relevant to good performance.

2. The relationships between cognitive capabilities can be complex and heterarchical, with single instances testing multiple cognitive capabilities with complex interactions. A carefully thought-out measurement layout is required, informed by cognitive science.

3. Instances may not be defined on all dimensions, introducing the possibility of missing data.

4. Instances are not necessarily evenly distributed across the levels of a dimension, leading to the potential for sampling biases.

### E.1 SYNTHETIC AGENTS AND COMPARISON

The synthetic agents Team A designed can be found in Table 5, Table 6, and Table 7.

### E.2 OBJECT PERMANENCE QUALITATIVE RESULTS

Figs. 19 and 20 show the models inferred values for each element of the cognitive profile. Fig 19 gives the mean of the inferred distribution and Fig .20 the standard deviation. These values were used by Team B to infer the qualitative type of agent being evaluated. Multiple aspects of the values inferred by the model need to be considered to make a proper assessment of each agent—particularly since Team B did not know a priori the exact types of agents that would exist. For example, Agent 2 has relatively low scores for most abilities, a high standard deviation for most abilities, and a value of $0.5$ for the mean and s.d of overall success. This led Team B to believe that this agent was succeeding the tasks at random with 50% probability. Equally, when looking at Agents 13 and 14, the right-left bias was high (with low standard deviation) leading Team B to believe that these agents were massively favouring movement in particular directions. We also present Forest Plots for agents, showing their 95% HDI across the 8 core capabilities in Figures 17 and 18.

In Fig. 21 we perform a qualitative comparison of the ground truth for the designed agents and the inferred profiles. The left side of the figure shows the ground truth characteristics of the agents generated by Team A. The right side of the figure shows Team B's conclusions about the characteristics of the agents based on the outputs of the model. The colours denote how well Team B's conclusions align with the ground truth, with green being more accurate and red being less accurate.

Using the measurement layout, Team B were able to recover all reference agents accurately. Team B were also able to recover the majority of the 'Achilles Heel' agents with high accuracy. Notable failures are the identification of a slight weakness with Lava for Agent 8 and a slight weakness with ramps for Agent 14. Agent 8 had a 90% probability of failing a task containing an occluder with a greenness value over 200, which has no bearing on the presence or absence of lava. It is possible that this small bias towards passing tasks without lava is a spurious result derived from the random values sampled to generate the performances. Agent 14 struggles with similar looking locations for

| Name | Agent Class | Agent Description |
|------|-------------|-------------------|
| Agent 1 | Reference | Passes every instance. |
| Agent 2 | Reference | Passes each instance with a probability of 0.5. |
| Agent 3 | Reference | Passes each instance with a probability of 0.1. |
| Agent 4 | Reference | Approximates a random walker. Passes a small proportion of instances. It is more likely to pass 'safe' instances that don't have lots of dangerous holes to fall down or lava to run into. |
| Agent 5 | Reference | Passes each instance with a probability of 0.9. This is a more human-like failure rate than agent 1. |
| Agent 6 | Achilles Heel | Has OP, passing an instance with a probability of 0.95 unless there is lava present, in which case it passes with a probability of 0.1. |
| Agent 7 | Achilles Heel | Has OP, passing an instance with a probability of 0.95 unless there is lava or an occluder with a 'redness' over 200 present, in which case it passes with a probability of 0.1. |
| Agent 8 | Achilles Heel | Has OP, passing an instance with a probability of 0.95 unless there is an occluder with a 'greenness' over 200 present, in which case it passes with a probability of 0.1. |
| Agent 9 | Achilles Heel | Has OP, passing an instance with a probability of 0.95 unless there is a ramp present, in which case it passes with a probability of 0.1. |
| Agent 10 | Achilles Heel | Has OP but lacks fine control of actions. If there is lava present, it only passes those instances with a probability of 0.5. It often falls into the wrong holes on its way to the further holes, so as distance to goal increases, the probability of passing Grid instances increases from 0.6 in increments of 0.1. This is only for the OP tasks. For the non-OP instances and the remaining instances, it solves an instance with a probability of 0.8. |

Table 5: The agents Team A designed and synthesised performances for. Agents 1-10.

reward locations, of which there are more for the grid tasks and the cup tasks, which both contain ramps by necessity, potentially explaining this ramp bias.

Team B were able to recover the cognitive profiles of Agents 17 and 18 to some degree. Team B inferred that Agent 17 struggled with memory, but because OP ability requires memory in the measurement layout, low memory ability was conflated with slightly lower OP. Agent 18 had worse performance when the reward could be hidden in more places. Similarly, the measurement layout stipulates that OP ability is required to track objects in multiple locations, so poorer performance on tasks with more hiding places was interpreted as a lower OP ability, a plausible inference.

Team B were able to partially recover the cognitive profiles of the context-specific agents. The relatively lower accuracy here was because task type was not included in the measurement layout. Therefore, many of the biases recovered by Team B reflect the particular construction of the paradigms used by Team A (see legend of Figure 21).

Team B did not accurately recover the cognitive profiles of the 'fraudster' agents. Both Agents 15 and 16 follow the policy of going to where the goal was last seen. This results in generally good performance in all tasks except the Chiandetti & Vallortigara tasks, where this policy would result in falling into lava. When this 'Fraudster' policy would be successful, these agents succeed with a probability of 0.95, if the policy would not be applicable, the agents succeeds with probability 0 (for agent 15) or 0.25 (for agent 16). Therefore, the performance distributions look similar to a high performing object permanence agent and our approach attributed higher values of OP than were warranted. To successfully discriminate these sophisticated agents, we would need to weight the importance of certain sub-groups of instances, such that, for example, Chiandetti & Vallortigara tasks are of most inferential importance in the case where there is good performance on the remain-

| Name | Agent Class | Agent Description |
|------|-------------|-------------------|
| Agent 11 | Achilles Heel | Has OP but poor visual acuity. If goals are 0.5 or smaller, they only pass the instance with a probability of 0.1. For goals of size 1, the probability is 0.3, it is 0.5 for size 1.5, 0.7 for size 2, 0.8 for size 2.5, and 0.9 for a goal size 3 or larger. |
| Agent 12 | Achilles Heel | Has OP but is biased to the right. Probability of passing a task is 0.95 when the goal is to the right, 0.3 when it is to the left, and 0.6 when it is centrally located. |
| Agent 13 | Achilles Heel | Has OP but is biased to the left. Probability of passing a task is 0.95 when the goal is to the left, 0.3 when it is to the right, and 0.6 when it is centrally located. |
| Agent 14 | Achilles Heel | Has OP, but is easily confused when there are several positions that are similar looking (either identical or mirror images of each other). |
| Agent 15 | Fraudster | Does not have OP, but goes to where they last saw the reward. Deterministically passes all instances where this policy works (all tasks except some of CV Chick tasks), and fails the rest. |
| Agent 16 | Fraudster | Does not have OP, but goes to where they last saw the reward. Passes all instances where this policy works with probability of 0.95 (all tasks except some of CV Chick tasks), and passes the rest with probability 0.25. |
| Agent 17 | Achilles Heel | Has OP, but has low memory, so struggles when the goal is occluded for longer. |
| Agent 18 | Achilles Heel | Has OP, but is easily confused when there are more positions that the goal could be occluded in. Passes instances with more than 10 other positions available with probability of 0.3. Between 5 and 10, probability of 0.6. Between 3 and 5, probability of 0.75. Less than 3, passes with probability of 0.95. |
| Agent 19 | Context-Specific | Non-robust OP. It can solve the navigation tasks and all the CV tasks, but fails the other instances. |
| Agent 20 | Context-Specific | Non-robust OP. It solves the navigation tasks and all the CV tasks with a probability of 0.75, fails the other paradigms with probability of 0.1. |

Table 6: The agents Team A designed and synthesised performances for. Agents 11-20.

ing paradigms. This requires good alignment between those designing the task and those creating the measurement layouts to make sure that controls established in the task design are reflected in the analysis.

Overall, however, Team B was able to extract a lot of information about a the majority of the synthetic agents—despite an imperfectly designed measurement layout. Also noteworthy is that when the measurement provides correct inferences, they tend to be consistent across that particular agent. That is, if the measurement layout identifies a particular weakness in an agent, it won't also pick a second, nonexistent weakness.

### E.3 OBJECT PERMANENCE PREDICTIVE RESULTS

The synthetic dataset was partitioned into an 80/20 train-test split. The measurement layout models (one for each agent) were trained on the same training partition. For each instance from the test-set we apply forward inference to yield a success probability.

We evaluate these probabilities using a Brier score (Brier, 1950), also making use of the decomposition into Calibration and Refinement (Murphy, 1973). Figure 5a plots Brier score against average success rate. The measurement layout model is consistently a better predictor of success than relying

| Name | Agent Class | Agent Description |
|---|---|---|
| Agent 21 | Context-Specific | Non-robust OP. It can solve the navigation tasks and all the Grid tasks, but fails the other instances. |
| Agent 22 | Context-Specific | Non-robust OP. It can solve the navigation tasks and all the Grid tasks with a probability of 0.75, fails the other paradigms with probability of 0.1. |
| Agent 23 | Context-Specific | Non-robust OP. It can solve the navigation tasks and all the 3 cup tasks, but fails the other instances. |
| Agent 24 | Context-Specific | Non-robust OP. It can solve the navigation tasks and all the 3 cup tasks with a probability of 0.75, fails the other paradigms with probability of 0.1. |
| Agent 25 | Context-Specific | Non-robust OP. It can solve the navigation tasks and all the CV and grid tasks, but fails the other instances. |
| Agent 26 | Cognitive Pathology | Non-robust OP. It can solve the navigation tasks and all the CV and grid tasks with a probability of 0.75, fails the other paradigms with probability of 0.1. |
| Agent 27 | Cognitive Pathology | Non-robust OP. It can solve the navigation tasks and all the 3 Cup and grid tasks, but fails the other instances. |
| Agent 28 | Cognitive Pathology | Non-robust OP. It can solve the navigation tasks and all the 3 cup and grid tasks with a probability of 0.75, fails the other paradigms with probability of 0.1. |
| Agent 29 | Cognitive Pathology | Non-robust OP. It can solve the navigation tasks and all the CV and 3 cup tasks, but fails the other instances. |
| Agent 30 | Cognitive Pathology | Non-robust OP. It can solve the navigation tasks and all the CV and 3 cup tasks with a probability of 0.75, fails the other paradigms with probability of 0.1. |

Table 7: The agents Team A designed and synthesised performances for. Agents 21-30.

| Capability | OP | FlatNav | Visual | Lava | Platform | Ramp | Memory |
|---|---|---|---|---|---|---|---|
| Overall RMSE | 0.28 | 0.18 | 0.16 | 0.20 | 0.28 | 0.38 | 0.15 |
| Included RMSE | 0.13 | 0.11 | 0.24 | 0.17 | 0.13 | 0.27 | 0.16 |

Table 8: RMSE between the ground truth cognitive profile for Team A's synthetic agents and the inferred values by Team B's measurement layout. Overall: All 30 agents. Included: 13 agents generated using only the features included in the model.

on the aggregate measure. We see better prediction performance for agents with less certain success rates. These are the agents for which we are more interested in improving prediction as there is more capacity to reduce uncertainty.

These scores are also given fully in Table 9, with the better performing approach for prediction in bold for each agent. As expected, the aggregate predictions have no refinement at all, and the values are close to 0.25 for balanced cases (the lower numbers only appear when the proportion of right/wrong performance is more imbalanced). The results from the measurement layout are much better here, as this method can give refined results per instance. When we look at calibration, however, we see aggregate predictions are almost perfect, since train and test come from the same distribution. But the calibration from the measurement layout is quite high, and overall there are many more cases where the measurement layout approach is superior than the aggregate. Moreover, it is important to emphasise that unlike this table, train and test do not have to come from the same distribution. For instance, if we removed the easy instances for many of the tasks, we would get similar results for the measurement layout prediction approach (calibration and refinement should

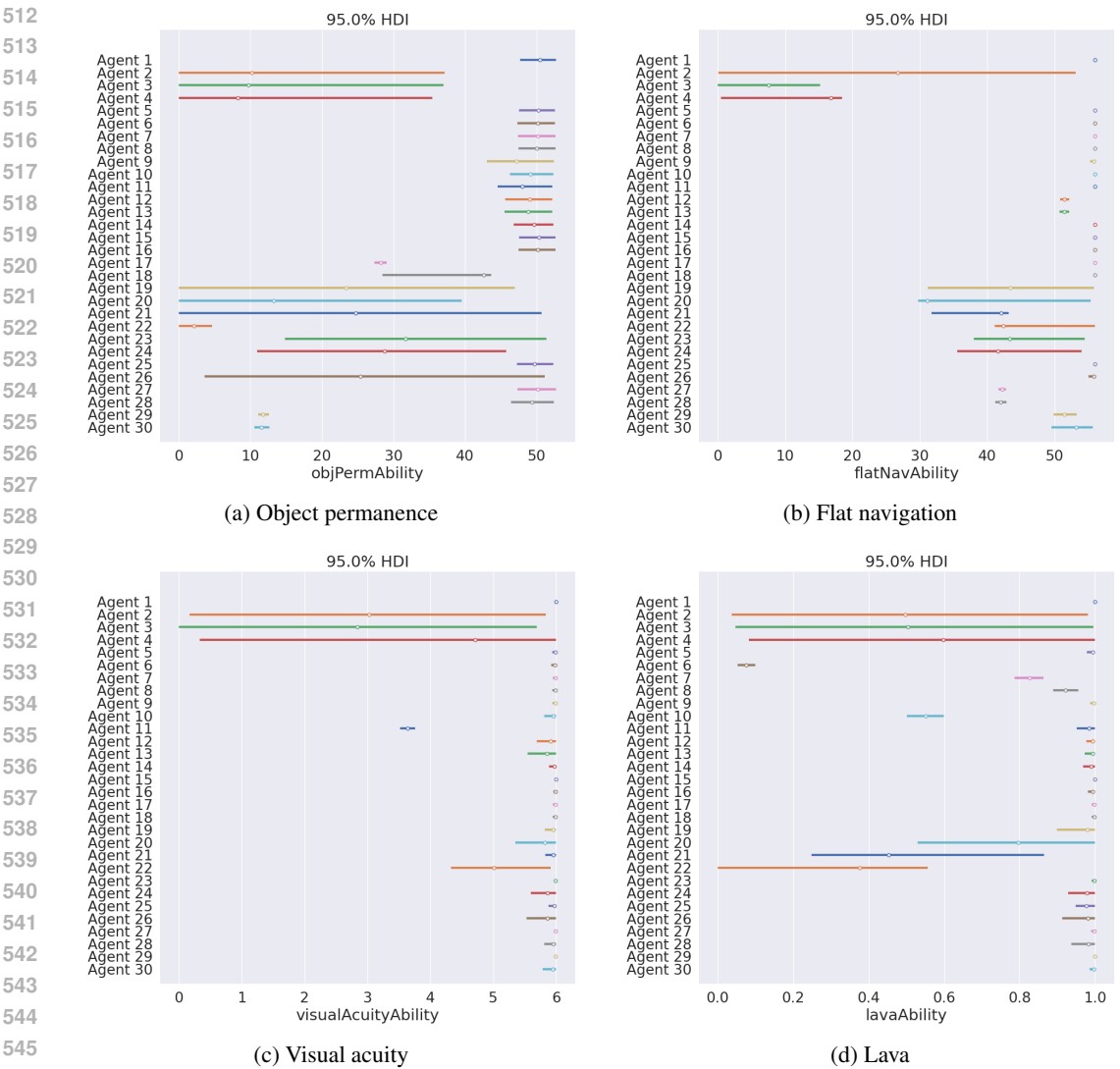

(a) Object permanence

(b) Flat navigation

(c) Visual acuity

(d) Lava

Figure 17: 95% HDIs for all agents across Object Permanence, Flat Navigation, Visual Acuity, and Lava.

not change dramatically), but the results for the aggregate prediction approach would be much worse, since the calibration would be completely lost. However, doing this

Looking at the particular agent properties that result in the measurement layouts to be more or less capable at prediction can provide some insight into the strengths and weaknesses of our approach.

Agents 1 and 5 were designed to pass all instances with probability 1 and 0.9 respectively. It's no surprise that predictions using the aggregate success rate were very successful. Our model's predictions were still highly accurate (Brier scores of 0.022 and 0.099 respectively), but the model lacked the extreme certainties of the aggregate measures. For Agent 1, this would be impossible— using Bayes' rule and an uninformed prior it's not possible to update the posterior to completely exclude one of the outcomes. For agent 5 we believe that our model didn't achieve the same level of certainty of success due to there not being sufficiently many instances to move the initial prior (uniform distribution) to the more extreme distribution predicting success at a rate of 0.9. Our model was able to cope with other fixed pass rates (such as with agents 2, 3, and 4), so this may just be a matter of degree. It's worth noting that for any fixed pass rate agent, it's actually impossible to predict more accurately than the aggregate in expectation. Hence for these types of agents the

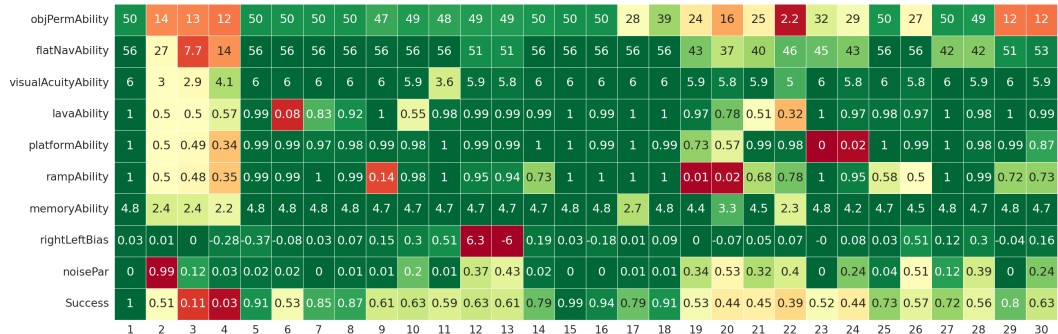

(a) Platform

(b) Ramp

(c) Right Left Bias

(d) Memory

Figure 18: 95% HDIs for all agents across Platforms, Ramps,

| | 1 | 2 | 3 | 4 | 5 | 6 | 7 | 8 | 9 | 10 | 11 | 12 | 13 | 14 | 15 | 16 | 17 | 18 | 19 | 20 | 21 | 22 | 23 | 24 | 25 | 26 | 27 | 28 | 29 | 30 |
|---|---|---|---|---|---|---|---|---|---|---|---|---|---|---|---|---|---|---|---|---|---|---|---|---|---|---|---|---|---|---|
| objPermAbility | 50 | 14 | 13 | 12 | 50 | 50 | 50 | 50 | 47 | 49 | 48 | 49 | 49 | 50 | 50 | 50 | 28 | 39 | 24 | 16 | 25 | 2.2 | 32 | 29 | 50 | 27 | 50 | 49 | 12 | 12 |
| flatNavAbility | 56 | 27 | 7.7 | 14 | 56 | 56 | 56 | 56 | 56 | 56 | 56 | 51 | 51 | 56 | 56 | 56 | 56 | 56 | 43 | 37 | 40 | 46 | 45 | 43 | 56 | 56 | 42 | 42 | 51 | 53 |
| visualAcuityAbility | 6 | 3 | 2.9 | 4.1 | 6 | 6 | 6 | 6 | 6 | 5.9 | 3.6 | 5.9 | 5.8 | 6 | 6 | 6 | 6 | 6 | 5.9 | 5.8 | 5.9 | 5 | 6 | 5.8 | 6 | 5.8 | 6 | 5.9 | 6 | 5.9 |
| lavaAbility | 1 | 0.5 | 0.5 | 0.57 | 0.99 | 0.08 | 0.83 | 0.92 | 1 | 0.55 | 0.98 | 0.99 | 0.99 | 0.99 | 1 | 0.99 | 1 | 1 | 0.97 | 0.78 | 0.51 | 0.32 | 1 | 0.97 | 0.98 | 0.97 | 1 | 0.98 | 1 | 0.99 |
| platformAbility | 1 | 0.5 | 0.49 | 0.34 | 0.99 | 0.99 | 0.97 | 0.98 | 0.99 | 0.98 | 1 | 0.99 | 0.99 | 1 | 1 | 0.99 | 1 | 0.99 | 0.73 | 0.57 | 0.99 | 0.98 | 0 | 0.02 | 1 | 0.99 | 1 | 0.98 | 0.99 | 0.87 |
| rampAbility | 1 | 0.5 | 0.48 | 0.35 | 0.99 | 0.99 | 1 | 0.99 | 0.14 | 0.98 | 1 | 0.95 | 0.94 | 0.73 | 1 | 1 | 1 | 1 | 0.01 | 0.02 | 0.68 | 0.78 | 1 | 0.95 | 0.58 | 0.5 | 1 | 0.99 | 0.72 | 0.73 |
| memoryAbility | 4.8 | 2.4 | 2.4 | 2.2 | 4.8 | 4.8 | 4.8 | 4.8 | 4.7 | 4.7 | 4.7 | 4.7 | 4.7 | 4.7 | 4.8 | 4.8 | 2.7 | 4.8 | 4.4 | 3.3 | 4.5 | 2.3 | 4.8 | 4.2 | 4.7 | 4.5 | 4.8 | 4.7 | 4.8 | 4.7 |
| rightLeftBias | 0.03 | 0.01 | 0 | -0.28 | -0.37 | -0.08 | 0.03 | 0.07 | 0.15 | 0.3 | 0.51 | 6.3 | -6 | 0.19 | 0.03 | -0.18 | 0.01 | 0.09 | 0 | -0.07 | 0.05 | 0.07 | -0 | 0.08 | 0.03 | 0.51 | 0.12 | 0.3 | -0.04 | 0.16 |
| noisePar | 0 | 0.99 | 0.12 | 0.03 | 0.02 | 0.02 | 0 | 0.01 | 0.01 | 0.2 | 0.01 | 0.37 | 0.43 | 0.02 | 0 | 0 | 0.01 | 0.01 | 0.34 | 0.53 | 0.32 | 0.4 | 0 | 0.24 | 0.04 | 0.51 | 0.12 | 0.39 | 0 | 0.24 |
| Success | 1 | 0.51 | 0.11 | 0.03 | 0.91 | 0.53 | 0.85 | 0.87 | 0.61 | 0.63 | 0.59 | 0.63 | 0.61 | 0.79 | 0.99 | 0.94 | 0.79 | 0.91 | 0.53 | 0.44 | 0.45 | 0.39 | 0.52 | 0.44 | 0.73 | 0.57 | 0.72 | 0.56 | 0.8 | 0.63 |

Figure 19: Mean inference results for the cognitive profile elements, along with the mean success rate, for the 30 synthetic agents in the OP task.

Aggregate Brier score is minimised, and for many fixed pass rate agents, our measurement layouts matched the aggregate's Brier score.

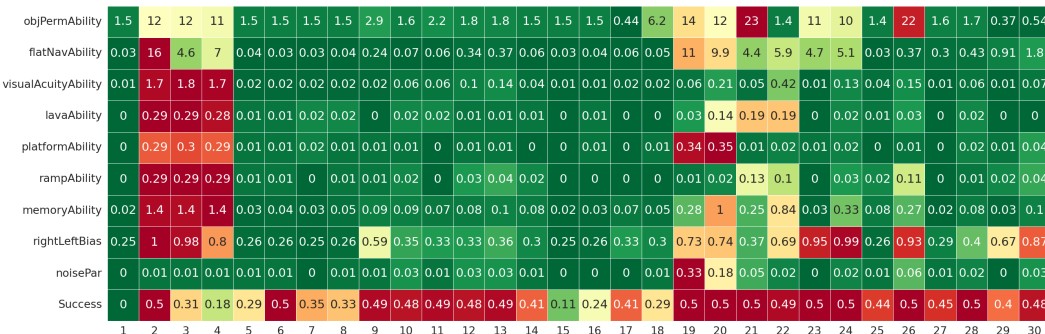

Figure 20: Standard Deviations of inferred cognitive profile elements, along with the standard deviation of the success rate, for the 30 synthetic agents in the OP task.

| Agent | Design | | | | | Model Decision | | | | |
|---|---|---|---|---|---|---|---|---|---|---|
| | Agent Type | Has OP? | Description | Weakness | Variability | Agent Type | Description | Weakness | Noise | Has OP? |
| Agent4 | Random | No | Random | Nonspecific / Reliability | Low | Reference | High Failure | Nonspecific | Low | No |
| Agent1 | Reference | Yes | Perfect | None | Low | Reference | Perfect | None | V. Low | Yes |
| Agent2 | | No Evidence | 50% Failure | Nonspecific | High | Random/ Reference | Random / 50% Performance | Nonspecific/Reliability | V. High | Low Confidence |
| Agent3 | | No | High Failure | Nonspecific | Low | Reference | High Failure | Nonspecific / Navigation on flat ground | Medium-Low | No |
| Agent5 | | Yes | High Performance | Nonspecific | Low | Reference | High Performance / Side Bias | Nonspecific / Slight Right Bias | Low | Yes |
| Agent6 | Achilles Heel | Yes | Navigation issues | Lava | Low | Achilles Heel | Navigation issues | Lava | Low | Yes |
| Agent7 | | Yes | Navigation issues | Lava + Occluders | Low | Achilles Heel | Navigation issues | Lava | V. Low | Yes |
| Agent8 | | Yes | Navigation issues | Occluders | Low | Achilles Heel | Navigation issues | (Slight) Lava | Low | Yes |
| Agent9 | | Yes | Navigation issues | Ramps | Low | Achilles Heel | Navigation issues | Ramp, Slight Right bias | V. Low | Yes |
| Agent10 | | Yes | Navigation issues (clumsy) | Platforms / Lava | High | Achilles Heel | Navigation issues | Lava | Medium | Low Confidence |
| Agent11 | | Yes | Vision Issues | Visual Acuity | Low | Achilles Heel | Vision Issues | Visual Actuiy / Slight Right Bias | Low | Yes |
| Agent12 | | Yes | Side Bias | Right Bias | Medium | Achilles Heel | Side Bias | Right Bias | Medium | Low Confidence |
| Agent13 | | Yes | Side Bias | Left Bias | Medium | Achilles Heel | Side Bias | Left Bias | Medium | Low Confidence |
| Agent14 | | Yes | Vision / Navigation Issues | Discriminating similar items | Low | Achilles Heel | Navigation Issues | (Slight) Ramp | Low | Yes |
| Agent17 | | Yes / Maybe | Lacks component ability | Memory | Medium | Low OP + Achilles Heel | Low OP + Memory Issues | (Slight) OP + Memory | Low | Maybe |
| Agent18 | | Yes / Maybe | Lacks component ability | Visual Search | Medium-low | Low OP | Low OP | (Slight) OP | Low | Maybe |
| Agent25 | Medium OP / Context-Specific | Yes/Maybe | Solves 2/3 types of OP tasks | The Cup OP Task* | Low | Achilles Heel | Navigation issues | Ramps | Low | Yes |
| Agent26 | | Yes/Maybe | Solves 2/3 types of OP tasks | The Cup OP Task | Medium | Low OP + Achilles Heel | Low OP + Navigation issues + Side Bias | OP + Ramps + Right Bias | High | Low Confidence |
| Agent27 | | Yes/Maybe | Solves 2/3 types of OP tasks | The CV OP Task** | Low | Achilles Heel | Navigations Issues | (Slight) Navigation on flat ground | Medium-Low | Low Confidence |
| Agent28 | | Yes/Maybe | Solves 2/3 types of OP tasks | The CV OP Task | Medium | Achilles Heel | Navigations Issues | (Slight) Navigation on flat ground + Right Bias | Medium | Low Confidence |
| Agent29 | | Yes/Maybe | Solves 2/3 types of OP tasks | The Grid OP Task*** | Low | Low OP + Achilles Heel | Low OP + Navigation issues | OP + Ramps | V. Low | No |
| Agent30 | | Yes/Maybe | Solves 2/3 types of OP tasks | The Grid OP Task | Medium | Low OP + Achilles Heel | Low OP + Navigation issues | OP + Ramps | Medium | Low Confidence |
| Agent19 | Low OP/ Highly Context-Specific | No/ Maybe | Solves 1/3 types of OP tasks | OP Tasks (not CV) | Low | Low OP + Achilles Heel | Low OP + Navigation issues | OP + Ramps | V. Low | No |
| Agent20 | | No/ Maybe | Solves 1/3 types of OP tasks | OP Tasks (not CV) | Medium | Low OP + Achilles Heel | Low OP + Navigation issues | OP + Ramps + Navigation on flat ground/ Platforms | Medium | Low Confidence |
| Agent21 | | No/ Maybe | Solves 1/3 types of OP tasks | OP Tasks (not Grid) | Low | Achilles Heel | Navigation issues | Lava + Navigation on flat ground + Ramps | Medium | Low Confidence |
| Agent22 | | No/ Maybe | Solves 1/3 types of OP tasks | OP Tasks (not Grid) | Medium | Low OP + Achilles Heel | Low OP + Memory/ Navigation Issues | OP + Memory + Lava + Navigation on flat ground | Medium | Low Confidence |
| Agent23 | | No/ Maybe | Solves 1/3 types of OP tasks | OP Tasks (not Cup) | Low | Low OP + Achilles Heel | Low OP + Navigation Issues | OP + Platforms | V. Low | Maybe |
| Agent24 | | No/ Maybe | Solves 1/3 types of OP tasks | OP Tasks (not Cup) | Medium | Low OP + Achilles Heel | Low OP + Navigation Issues | OP + Platforms | Medium | Low Confidence |
| Agent15 | Fraudster | No | Fraudster (Uses Strategy) | OP | Low | Reference | High Performance | None/Nonspecific | V. Low | Yes |
| Agent16 | Fraudster | No | Fraudster (Uses Strategy) | OP | Low | Achilles Heel | Side Bias | (Slight) Left Bias | V. Low | Yes |

Figure 21: Qualitative analysis of the 30 synthetic agents, comparing the characteristics as they were designed with the cognitive profile that we have identified with our model decision for the measurement layout. In a gradient from green-yellow-red, with green representing a successful identification and red an unsuccessful one.

Agents 7 and 8 were predicted equally or better by the Aggregate measure than our model. These agents passed all instances with probability 0.95, unless the object was occluded by an object with a lot of red or green present in its RGB colour value (red for 7, green for 8). Given that the measurement layout did not include colour values, it's easy to see why they would struggle to predict this complex behaviour accurately. On the other hand, they did not do much worse than the aggregate prediction, with Brier scores of 0.138 vs 0.138 and 0.109 vs 0.103 respectively.

The only other agents that our measurement layouts were worse at predicting than the aggregate were agents 15 and 16. These two agents were designed as 'Fraudster' agents that mimic object permanence by going to the last position where they saw the reward. We discussed these agents and their strategy in the Section 4 about Qualitative Validity. Since these agents had very high overall

| Agent | ML BS | Agg. BS | ML Cal. | Agg. Cal. | ML Ref. | Agg. Ref. |
|-------|-------|---------|---------|-----------|---------|-----------|
| 1  | 0.022 | **0.000** | 0.022 | 0.000 | 0.000 | 0.000 |
| 2  | **0.250** | **0.250** | 0.000 | 0.000 | 0.250 | 0.250 |
| 3  | **0.078** | **0.078** | 0.001 | 0.001 | 0.077 | 0.077 |
| 4  | **0.022** | **0.022** | 0.000 | 0.000 | 0.022 | 0.022 |
| 5  | 0.099 | **0.089** | 0.011 | 0.000 | 0.087 | 0.089 |
| 6  | **0.067** | 0.250 | 0.011 | 0.000 | 0.056 | 0.250 |
| 7  | **0.138** | **0.138** | 0.012 | 0.000 | 0.125 | 0.137 |
| 8  | 0.109 | **0.103** | 0.008 | 0.000 | 0.101 | 0.103 |
| 9  | **0.060** | 0.240 | 0.005 | 0.000 | 0.055 | 0.239 |
| 10 | **0.211** | 0.237 | 0.002 | 0.000 | 0.209 | 0.236 |
| 11 | **0.176** | 0.242 | 0.012 | 0.000 | 0.164 | 0.242 |
| 12 | **0.218** | 0.235 | 0.003 | 0.000 | 0.215 | 0.235 |
| 13 | **0.222** | 0.234 | 0.004 | 0.001 | 0.218 | 0.234 |
| 14 | **0.152** | 0.166 | 0.008 | 0.000 | 0.145 | 0.166 |
| 15 | 0.026 | **0.007** | 0.020 | 0.000 | 0.007 | 0.007 |
| 16 | 0.077 | **0.064** | 0.012 | 0.000 | 0.063 | 0.064 |
| 17 | **0.090** | 0.166 | 0.022 | 0.000 | 0.068 | 0.166 |
| 18 | **0.069** | 0.072 | 0.008 | 0.000 | 0.061 | 0.072 |
| 19 | **0.056** | 0.249 | 0.004 | 0.000 | 0.051 | 0.249 |
| 20 | **0.177** | 0.246 | 0.016 | 0.000 | 0.162 | 0.246 |
| 21 | **0.156** | 0.249 | 0.023 | 0.001 | 0.132 | 0.248 |
| 22 | **0.204** | 0.248 | 0.013 | 0.004 | 0.191 | 0.245 |
| 23 | **0.012** | 0.250 | 0.011 | 0.003 | 0.000 | 0.247 |
| 24 | **0.165** | 0.245 | 0.005 | 0.000 | 0.159 | 0.245 |
| 25 | **0.149** | 0.204 | 0.023 | 0.000 | 0.125 | 0.204 |
| 26 | **0.229** | 0.248 | 0.012 | 0.001 | 0.217 | 0.247 |
| 27 | **0.084** | 0.192 | 0.034 | 0.002 | 0.048 | 0.190 |
| 28 | **0.186** | 0.242 | 0.028 | 0.003 | 0.158 | 0.239 |
| 29 | **0.098** | 0.152 | 0.056 | 0.000 | 0.040 | 0.152 |
| 30 | **0.205** | 0.232 | 0.013 | 0.000 | 0.191 | 0.232 |

Table 9: Brier Scores (BS) and the breakdown into calibration (Cal.) and refinement (Ref.) for the Measurement Layouts (ML) and aggregate predictions (Agg.). Bold indicates whether the Measurement Layout or Aggregate Brier score was lower (and therefore better).

success rates, we believe that this causes our model to be less predictive, similar to Agent 5 (and to a lesser extent Agent 7 and 8), because it takes more task instances to move an uninformative policy to extremes such as predicting success with a probability greater than $0.9$.

# F  SUPPLEMENTARY EXPERIMENT: ANIMAL AI OLYMPICS

This experiment exemplifies that evaluation suites that weren't designed for this type of nuanced analysis can still provide considerable insights.

The Animal AI Olympics (AAIO) was an open competition evaluating AI agents in the AAI 3D environment, where agents had to get a reward by locating it and navigating in an arena. In the Animal-AI Olympics (Crosby et al., 2020), an example arena containing the possible objects was provided to contestants. They were instructed that their agents would be evaluated on a variety of problems constructed from the objects in the example arena. The contestants then needed to build an agent—using any method of their choosing—that they believed would succeed at a wide variety of tasks. The tasks that ultimately made up the evaluation suite were based on adaptations of tasks from comparative and developmental literature. More information can be found at `http://animalai.org/animal-ai-olympics`. For our work with the AAIO dataset we used a subset of 69 tasks instances that were solvable with simple goal-directed behaviour. The exact instances can be found in Appendix F.1. We include 68 participant agents for which we know nothing about their design.

### F.1 ANIMAL AI OLYMPICS INSTANCES

For our work with the AAIO dataset we used a subset of tasks instances that were solvable with simple goal-directed behaviour. The specific task instances we used were:

[1-1-1, 1-1-2, 1-1-3, 1-2-1, 1-2-2, 1-2-3, 1-3-1, 1-3-2, 1-3-3, 1-4-1, 1-4-2, 1-4-3, 1-5-1, 1-5-2, 1-5-3, 1-6-1, 1-6-2, 1-6-3, 1-7-1, 1-7-2, 1-7-3, 1-8-1, 1-8-2, 1-8-3, 1-9-1, 1-9-2, 1-9-3, 1-10-1, 1-10-2, 1-10-3, 1-11-1, 1-11-2, 1-11-3, 1-12-1, 1-12-2, 1-12-3, 1-13-1, 1-13-2, 1-13-3, 1-14-1, 1-14-2, 1-14-3, 1-15-1, 1-15-2, 1-15-3, 1-16-1, 1-16-2, 1-16-3, 1-17-1, 1-17-2, 1-17-3, 7-1-1, 7-1-2, 7-1-3, 7-2-1, 7-2-2, 7-2-3, 7-3-1, 7-3-2, 7-3-3, 7-4-1, 7-4-2, 7-4-3, 7-5-1, 7-5-2, 7-5-3, 7-6-1, 7-6-2, 7-6-3]

The types of arenas that these instances correspond to can be found at (`http://animalai.org/animal-ai-olympics`).

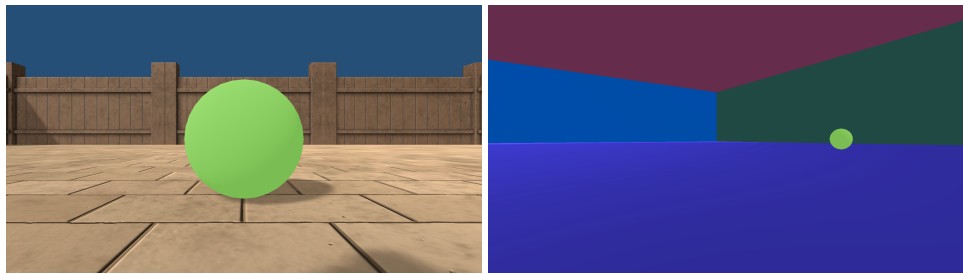

Figure 22: Screenshots from two different instances in the Animal AI Olympics that we include in our analysis.

| Dimension | Description | Range/Values |
|---|---|---|
| Reward Size | The size of the reward (the sign is modified so lower is larger) | $[0, 1.9]$ |
| Reward Distance | Euclidean distance from the agent's start position to the reward | $[0, 5.3]$ |
| Reward Behind | Whether the reward originates behind (1), left or right (0.5) or in front (0) of the agent | $\{0, 0.5, 1\}$ |
| XPos | Whether the reward is to the left ($-1$), the right (1) or is centred (0) relative to the agent's start position | $\{-1, 0, 1\}$ |

Table 10: AAIO Task Characterisation details. The dimensions on which instances vary in our measurement layouts.

Note that these ranges determine the ranges of the capabilities, so that we have units. For instance, if distance to the reward were measured in metres then the navigation capability would have metres as a unit. Let the navigation capability of an agent be, say, 20 metres. Since it is defined with a logistic function of distance, that means that the agent can achieve excellent performance when well below 20, performing at chance around the 20 metre mark, and failing every instance when the reward is well beyond 20 metres away.

Fig. 23 shows the complete inference results for the 68 agents in the AAIO. The agents are ordered by the aggregate success rate at the collection of task instances. We see that, for the agents in the middle of the table in particular, our approach is able to decompose agent capabilities at this task into the constituent navigation and visual abilities. Bias has a more diverse behaviour throughout the table, with strong biases for extremely low scoring agents too, such as with agents 'ahorizon' and 'Octopus'.

Equally, we can see that the detected bias can account for poor performance. Compare 'Thursday' and 'MadStorks' which share a similar overall success rate (0.30 vs. 0.28) despite the fact that 'Thursday' has much more navigation ability than 'MadStorks' and they have similar visual abilities.

| | navigationAbility | | visualAbility | | noiseLevel | | rightLeftBias | | Success | |
|---|---|---|---|---|---|---|---|---|---|---|
| | mean | stddev | mean | stddev | mean | stddev | mean | stddev | mean | stddev |
| Oltau.ai | 4.77 | 0.41 | 1.85 | 0.08 | 0.02 | 0.02 | -0.14 | 0.84 | 0.99 | 0.12 |
| Psidon | 4.75 | 0.40 | 1.85 | 0.08 | 0.02 | 0.01 | -0.16 | 0.83 | 0.99 | 0.12 |
| Trrrrr | 4.79 | 0.41 | 1.85 | 0.08 | 0.02 | 0.01 | -0.13 | 0.86 | 0.99 | 0.12 |
| ironbar | 4.88 | 0.34 | 1.84 | 0.08 | 0.01 | 0.01 | -0.14 | 0.84 | 0.99 | 0.12 |
| sirius | 4.87 | 0.35 | 1.85 | 0.08 | 0.01 | 0.01 | -0.13 | 0.86 | 0.99 | 0.12 |
| oreleus | 4.73 | 0.44 | 1.84 | 0.08 | 0.02 | 0.02 | -0.24 | 0.91 | 0.96 | 0.20 |
| Melflo | 4.60 | 0.49 | 1.84 | 0.08 | 0.03 | 0.03 | 0.19 | 0.79 | 0.94 | 0.23 |
| Neo | 4.69 | 0.45 | 1.83 | 0.08 | 0.02 | 0.02 | -0.26 | 0.87 | 0.94 | 0.23 |
| winter2109 | 4.73 | 0.43 | 1.83 | 0.09 | 0.02 | 0.02 | -0.02 | 0.86 | 0.94 | 0.23 |
| DeepFox | 4.64 | 0.46 | 1.82 | 0.10 | 0.03 | 0.02 | 0.1 | 0.90 | 0.93 | 0.26 |
| mmIA | 4.78 | 0.42 | 1.83 | 0.10 | 0.04 | 0.03 | -0.32 | 0.91 | 0.93 | 0.26 |
| sungbinchoi | 4.76 | 0.42 | 1.82 | 0.10 | 0.03 | 0.03 | -0.36 | 0.94 | 0.93 | 0.26 |
| crazy animals | 4.43 | 0.55 | 1.83 | 0.09 | 0.03 | 0.03 | 0.17 | 0.85 | 0.91 | 0.28 |
| BronzeBlood | 4.32 | 0.61 | 1.83 | 0.09 | 0.05 | 0.04 | 0.08 | 0.90 | 0.90 | 0.30 |
| Juramaia | 4.64 | 0.51 | 1.81 | 0.10 | 0.04 | 0.03 | -0.03 | 0.88 | 0.90 | 0.30 |
| ARF-RL | 4.63 | 0.49 | 1.80 | 0.11 | 0.04 | 0.03 | 0.33 | 0.96 | 0.88 | 0.32 |
| Bonum | 4.29 | 0.62 | 1.80 | 0.11 | 0.03 | 0.02 | -0.16 | 0.84 | 0.88 | 0.32 |
| 41Animals | 4.37 | 0.59 | 1.81 | 0.11 | 0.05 | 0.04 | 0.45 | 0.91 | 0.87 | 0.34 |
| cso | 4.67 | 0.48 | 1.78 | 0.13 | 0.06 | 0.04 | -0.15 | 0.91 | 0.86 | 0.35 |
| BABL AI | 4.34 | 0.61 | 1.76 | 0.14 | 0.05 | 0.04 | -0.22 | 0.95 | 0.84 | 0.37 |
| Gyutan | 4.19 | 0.65 | 1.78 | 0.13 | 0.07 | 0.05 | -0.04 | 0.89 | 0.83 | 0.38 |
| animalAI-challenge | 4.15 | 0.65 | 1.76 | 0.14 | 0.05 | 0.04 | 0.03 | 0.83 | 0.83 | 0.38 |
| CUMIN | 4.12 | 0.66 | 1.77 | 0.14 | 0.07 | 0.05 | 0.31 | 0.96 | 0.81 | 0.39 |
| inf mnky | 4.51 | 0.56 | 1.74 | 0.16 | 0.11 | 0.06 | 0.15 | 0.97 | 0.80 | 0.40 |
| CHROMA | 3.84 | 0.74 | 1.72 | 0.17 | 0.06 | 0.05 | 0.5 | 0.91 | 0.78 | 0.41 |
| UniboTeam | 2.94 | 0.63 | 1.79 | 0.12 | 0.06 | 0.05 | 0.11 | 0.82 | 0.78 | 0.41 |
| doot | 4.02 | 0.69 | 1.73 | 0.16 | 0.08 | 0.06 | 0.22 | 0.89 | 0.78 | 0.41 |
| forest | 3.20 | 0.77 | 1.79 | 0.12 | 0.10 | 0.07 | 0.22 | 0.89 | 0.77 | 0.42 |
| GoGoAI | 4.01 | 0.75 | 1.73 | 0.17 | 0.14 | 0.08 | 0.16 | 0.89 | 0.75 | 0.43 |
| Horsepower | 2.94 | 0.74 | 1.71 | 0.18 | 0.09 | 0.07 | 0.49 | 0.80 | 0.72 | 0.45 |
| Qodiak | 4.15 | 0.74 | 1.63 | 0.22 | 0.15 | 0.08 | 0.46 | 1.06 | 0.72 | 0.45 |
| INAOE1 | 2.44 | 0.60 | 1.74 | 0.15 | 0.12 | 0.09 | 0.44 | 0.81 | 0.70 | 0.46 |
| y.yang | 3.82 | 0.85 | 1.68 | 0.21 | 0.22 | 0.11 | 0.25 | 0.98 | 0.70 | 0.46 |
| BLAI | 3.90 | 0.88 | 1.73 | 0.17 | 0.34 | 0.11 | 0.01 | 0.87 | 0.68 | 0.47 |
| KMU-AIL | 3.55 | 0.93 | 1.52 | 0.30 | 0.29 | 0.14 | 0.05 | 0.88 | 0.64 | 0.48 |
| Redstone Blockchain AI | 1.84 | 0.68 | 1.55 | 0.30 | 0.20 | 0.15 | 0.52 | 0.65 | 0.58 | 0.49 |
| jinrohs | 2.98 | 1.10 | 1.02 | 0.42 | 0.25 | 0.16 | -0.73 | 1.03 | 0.57 | 0.50 |
| AirbrainNM | 1.50 | 0.70 | 1.32 | 0.40 | 0.22 | 0.16 | 0.23 | 0.80 | 0.54 | 0.50 |
| Juohmaru | 2.50 | 1.08 | 0.92 | 0.44 | 0.26 | 0.17 | 0.14 | 0.85 | 0.54 | 0.50 |
| daydayup | 3.48 | 1.26 | 0.74 | 0.46 | 0.48 | 0.20 | 0.08 | 0.98 | 0.52 | 0.50 |
| Koozyt_Hiperdyne | 1.97 | 1.02 | 0.80 | 0.46 | 0.29 | 0.19 | -0.78 | 0.98 | 0.49 | 0.50 |
| Araya | 1.92 | 1.31 | 0.55 | 0.41 | 0.41 | 0.19 | 0.02 | 0.89 | 0.45 | 0.50 |
| TheAnimalsEscapedFromPenn | 2.18 | 1.00 | 0.44 | 0.32 | 0.16 | 0.13 | -0.24 | 0.95 | 0.45 | 0.50 |
| Koozyt_AnimalAI_alpha | 1.05 | 0.60 | 0.65 | 0.42 | 0.20 | 0.09 | 0.45 | 0.81 | 0.41 | 0.49 |
| ocorcoll-AI | 1.65 | 0.90 | 0.31 | 0.27 | 0.13 | 0.11 | 0.4 | 0.95 | 0.39 | 0.49 |
| cocel | 0.25 | 0.24 | 1.07 | 0.51 | 0.26 | 0.13 | 0.58 | 0.69 | 0.36 | 0.48 |
| sparklemotion | 0.40 | 0.38 | 0.59 | 0.45 | 0.31 | 0.13 | 0.01 | 0.81 | 0.36 | 0.48 |
| fep-bot | 0.67 | 0.49 | 0.38 | 0.31 | 0.08 | 0.07 | 0.4 | 0.71 | 0.32 | 0.47 |
| Thursday | 0.71 | 0.50 | 0.34 | 0.29 | 0.06 | 0.06 | 1.14 | 0.76 | 0.30 | 0.46 |
| MadStorks | 0.39 | 0.33 | 0.32 | 0.27 | 0.08 | 0.06 | 0.19 | 0.79 | 0.28 | 0.45 |
| ACCESS | 0.37 | 0.31 | 0.30 | 0.26 | 0.09 | 0.06 | 0.68 | 0.69 | 0.26 | 0.44 |
| NARUTO | 0.37 | 0.31 | 0.30 | 0.26 | 0.08 | 0.06 | 0.69 | 0.72 | 0.26 | 0.44 |
| Optimize Prime Squad | 0.34 | 0.31 | 0.33 | 0.28 | 0.08 | 0.06 | 0.71 | 0.71 | 0.26 | 0.44 |
| Yossy | 0.36 | 0.32 | 0.31 | 0.27 | 0.08 | 0.06 | 0.66 | 0.74 | 0.26 | 0.44 |
| Yuko Ishizaki | 0.35 | 0.32 | 0.31 | 0.26 | 0.08 | 0.06 | 0.68 | 0.72 | 0.26 | 0.44 |
| bamasa_team | 0.35 | 0.31 | 0.31 | 0.27 | 0.08 | 0.06 | 0.65 | 0.73 | 0.26 | 0.44 |
| bird | 0.35 | 0.32 | 0.31 | 0.27 | 0.08 | 0.06 | 0.67 | 0.71 | 0.26 | 0.44 |
| dhr | 0.36 | 0.31 | 0.30 | 0.26 | 0.08 | 0.06 | 0.69 | 0.69 | 0.26 | 0.44 |
| hhq126152 | 0.36 | 0.32 | 0.32 | 0.27 | 0.08 | 0.06 | 0.65 | 0.74 | 0.26 | 0.44 |
| loneWolf | 0.36 | 0.31 | 0.30 | 0.27 | 0.08 | 0.06 | 0.68 | 0.72 | 0.26 | 0.44 |
| vithng | 0.37 | 0.32 | 0.30 | 0.27 | 0.08 | 0.06 | 0.69 | 0.73 | 0.26 | 0.44 |
| Tetravoxel | 0.25 | 0.23 | 0.39 | 0.31 | 0.07 | 0.06 | -0.32 | 0.64 | 0.25 | 0.43 |
| ahorizon | 0.28 | 0.25 | 0.26 | 0.24 | 0.12 | 0.07 | 0.63 | 0.71 | 0.25 | 0.43 |
| aSphericalChicken | 0.33 | 0.30 | 0.22 | 0.20 | 0.08 | 0.05 | 0.73 | 0.72 | 0.23 | 0.42 |
| Octopus | 0.19 | 0.18 | 0.39 | 0.31 | 0.04 | 0.04 | -0.08 | 0.65 | 0.22 | 0.41 |
| Animal AI Team | 0.12 | 0.12 | 0.16 | 0.16 | 0.02 | 0.02 | 0.27 | 0.78 | 0.10 | 0.30 |
| Nishi-Hashi | 0.10 | 0.09 | 0.14 | 0.14 | 0.02 | 0.02 | 0.26 | 0.74 | 0.04 | 0.20 |
| ice-play | 0.10 | 0.10 | 0.13 | 0.12 | 0.02 | 0.02 | 0.26 | 0.74 | 0.04 | 0.20 |

Figure 23: Inferred cognitive profile elements, along with the success rate, for the 68 agents in the AAIO. The gradient colour red-yellow-green denotes how high the means are (green is best) while the gradient circles white-grey-black denote how high the standard deviations are (black is highest).

This is explained by 'Thursday''s much larger bias towards moving to the right (0.73 vs 0.4) even if the reward is to the left.

## F.2 COGNITIVE PROFILES

To illustrate how a measurement layout identifies cognitive profiles, we explore a range of tasks in the Animal-AI Environment (Beyret et al., 2019; ?). Full details of the environment can be found in Appendix B. we first select a simple navigation task where agents must navigate to a goal. In total, there are 69 selected navigation tasks and screenshots for two tasks are shown in Fig. 22. We include 68 agents submitted to the AAI Olympics competition (Crosby et al., 2020). The profiles of

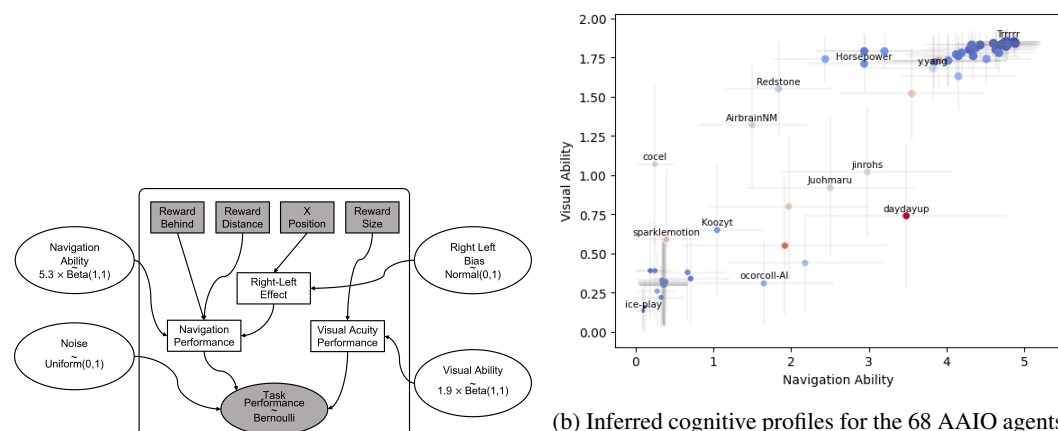

(a) Measurement layout for the Animal-AI Olympics.

(b) Inferred cognitive profiles for the 68 AAIO agents. Error bars show s.d.; dot area encodes mean accuracy; colour encodes noiseLevel (red=high).

Figure 24

agents will be composed of navigation and perception skills and whether they have any bias towards goals occuring on the left vs. the right.

Following the analysis performed in (Burnell et al., 2022), we selected the set of meta-features $X$ to characterise task instances as rewardSize, rewardDistance, Xpos and rewardBehind. These represent the reward's size, distance, and if it is left, right, in front, or behind the agent. Full details of the ranges of these variables and precise descriptions are presented in Appendix B, Table 10.

The cognitive profile is composed of two capabilities, navigationAbility and visualAbility, respectively representing the skill to move around purposefully and the visual acuity of the agent, one bias rightLeftBias, representing any preference for left or right movements, and one robustness level noiseLevel, accounting for any unexplained variability.

We then connect the task characterisation to the cognitive profile using a Measurement Layouts, that captures the interaction between demands and capabilities/bias. Fig. 24a presents the Measurement Layout we built. The metafeatures $X$ appear as observed (shaded, rectangular) nodes in the model, appearing within the plate, with 69 instances. The elements of the cognitive profile appear as external latent (unshaded) variables. The derived nodes lie at the intersection of the meta-features and the external cognitive profile nodes.

Not visible in the figure are the linking functions associated with each arrow. Broadly, the relevant metafeature task demands combine to increase difficulty, and are opposed by relevant capabilities, yielding partial performance estimates in the forms of probabilities. In this task, because success requires both identidying the goal and navigating too it, the partial performances are multiplied to calculate the final probability of success. Note that the ranges of these two abilities are given by the ranges of the task metafeatures, with navigationAbility in $[0, 5.3]$ and with visualAbility in $[0, 1.9]$ respectively, representing how distant and how big a reward—with their corresponding distance and size units—the agent could be measured as having in this task. This provides an intuitive property to the Measurement Layouts, in that the capabilities have meaningful units.

We detail all of the nodes and linking functions comprising this Measurement Layout in Table 11. The function $\sigma$ represents the standard logistic function, while $\omega$ represents a weighting:

$$\omega[\alpha](b, c) = (1 - \alpha) \cdot b + \alpha \cdot c$$

The value $\nu$ represents a noise prior that represents a constant model set to $1 -$ average performance. Finally, $ScaledBeta$ represents a scaled beta distribution to an interval different from $[0, 1]$. Note that a beta distribution has two parameters that need to be estimated. We describe our choices for building the measurement Layout in this way in Appendix F.

| Name | Type | Range/Values | Prior / Linking Function |
|------|------|--------------|--------------------------|
| rewardSize | Meta-Feature | $[0, 1.9]$ | |
| rewardDistance | Meta-Feature | $[0, 5.3]$ | |
| rewardBehind | Meta-Feature | $\{0, 0.5, 1\}$ | |
| XPos | Meta-Feature | $\{-1, 0, 1\}$ | |
| navigationAbility | Capability | $[0, 5.3]$ | $ScaledBeta(1, 1, 0, 5.3)$ |
| visualAbility | Capability | $[0, 1.9]$ | $ScaledBeta(1, 1, 0, 1.9)$ |
| rightleftBias | Bias | $[-\infty, \infty]$ | $\mathcal{N}(0, 1)$ |
| noiseLevel | Robustness | $[0, 1]$ | $Uniform(0, 1)$ |
| rightleftEffect | INON | $[-1, 1]$ | rightleftBias $\times$ XPos |
| navigationPerformance | INON | $[0, 1]$ | $\sigma$(NavigationAbility $-$ rewardDistance $\times$ ($\frac{1}{2}$rewardBehind $+ 1$) $+$ rightleftEffect) |
| visualPerformance | INON | $[0, 1]$ | $\sigma$(visualAbility $-$ rewardSize) |
| taskPerformance | Observed | $[0, 1]$ | $p \sim Bernoulli(\omega[\text{noiseLevel}](\text{navigationPerformance} \times \text{visualPerformance}, \nu))$ |

Table 11: Nodes in the AAIO Measurement Layout, including range/values, priors and linking function.

### F.3 INFERENCE AND ANALYSIS

Using the measurement layout and the 69 instances, we can use PyMC to infer the values of the cognitive profile for each of the 68 agents.

Fig. 24b summarises the inferred capabilities and noise levels. We see the two main capabilities are correlated for this population, however, we are also able to identify agents with weak navigation skill, but strong visual acuity (e.g.,cocel) and vice versa. We can also identify agents with strong biases (e.g., 'Thursday' and 'KoozytHiperdyne' with bias $1.14$ and $-0.78$). A full table with all of the inferred results for each agent is presented in Appendix F

Where most evaluations end, with a results table of identified scores, Measurement Layouts are novel insofar as they can go on to predict agent performance on unseen tasks instances. This happens by the forwards inference process of the HBN, using the To evaluate this, we randomly held out 20% of the task instances from the backwards inference stage and evaluated the predicted probabilities of success on them. We evaluate these probabilities using a Brier score (Brier, 1950). That is:

$$\text{Brier Score}(p, o) = \frac{1}{N} \sum_{i=1}^{N} (p_i - o_i)^2$$

Where $p_i$ and $o_i$ are the predicted probabilities of success and observed outcomes respectively. The Brier Score is essentially the mean square error applied to probabilities and binary events.

Due to the small size of the dataset, we repeated this process 15 times and took the mean Brier score for each agent. Figure 25 summarises these scores, which are lower than predictions based on aggregate performance for agents with success rates in the range $[0.2, 0.7]$, exactly where there is variability to explain.

### F.4 AAIO PATTERNS OF PERFORMANCE

To show the patterns of performance of several agents, we use a simple visualisation known as *agent characteristic grids*, as described in (Burnell et al., 2022). The plots place each feature as a dimension and the success rate represented in colours from green (success) to red (failure). Values are grouped into bins of appropriate width in the desired dimensions for analysis. Each cell also shows the number of observed instances in that bin. Fig. 26 shows some of the agents discussed in the paper, as well as some other agents that will serve us to illustrate a few points.

For instance, 'ironbar' has 99% accuracy (the grid is mostly green everywhere) for this set of tasks (like 'Trrrr', the winner of the competition) and is identified as saturating both capabilities in Fig. 24b. The next one, 'sparklemotion', with 36% accuracy, is very non-conformant, with navigationAbility $0.40 \pm 0.38$ and visualAbility $0.59 \pm 0.45$, and high noise $0.31 \pm 0.13$. Proportionally, navigationAbility is worse as the grid looks even less conformant on the distance of the reward than the reward size. The rightmost on the top row is 'Juohmaru', with 54% accuracy, which is a very conformant agent, with red concentrating on the top and right of the grid (the objectively most difficult instances). Its navigation ability is $2.50 \pm 1.08$ and visual ability is $0.92 \pm 0.44$, and medium noise: $0.26 \pm 0.17$.

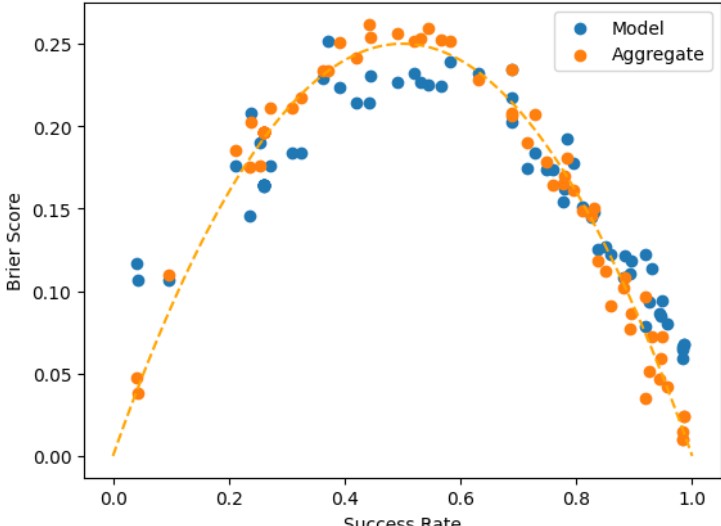

Figure 25: Brier score of model predictions against agent success rates for AAIO. The orange parabola is the expected value for the Brier score on aggregate predictions $BS = s(1 - s)$ for success rate $s$. The orange points are the observed Brier Scores for each agent using the aggregate prediction. These fall on the parabola as expected. The blue points are the Brier scores for our predictions using the Measurement Layout.

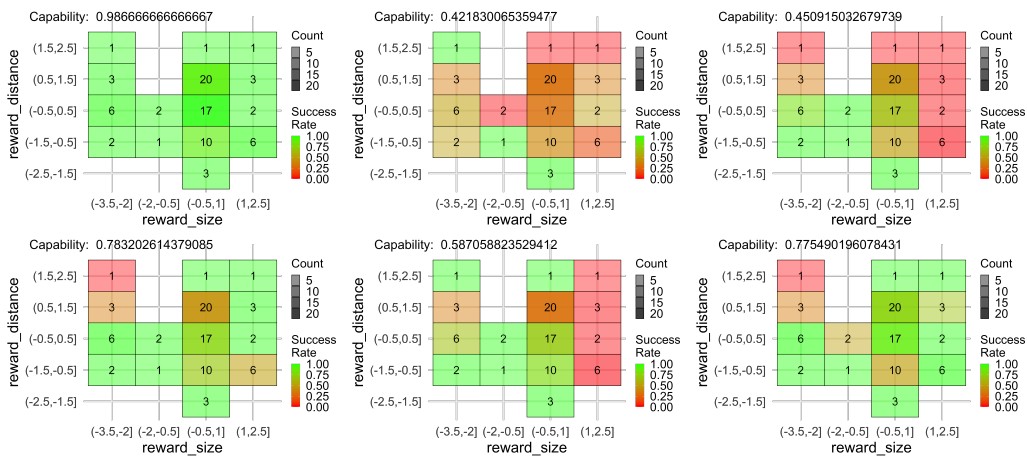

Figure 26: Agent characteristic grid displaying results for some selected agents. Top row, from left to right: 'ironbar', 'sparklemotion', 'Juohmaru'. Bottom row, from left to right: 'y.yang', 'daydayup', '41Animals'. Note that the scales of the $x$-axis and $y$-axis are normalised and not the same as in the main body of text.

We have specifically chosen some particular agents in the second row for which the extracted cognitive profile does not capture the whole performance patterns completely. Leftmost on the second row, 'y-yang' has 70% accuracy and a very strange grid, with red cells in a diagonal, indicating a very non-conformant agent. This translates into navigationAbility $3.82 \pm 0.45$ and visualAbility $1.68 \pm 0.21$, but medium noise $0.22 \pm 0.17$. This is so because overall the agent has high accuracy, so a bit of noise can explain the few anomalies on the top right of the grid. In the middle of the second row is 'daydayup' with 52% accuracy. This agent shows a higher navigation ability than visual ability, as can be seen in the grid, where the size of the reward is the predictive feature and reward distance is less relevant. This translates into navigationAbility $3.48 \pm 1.28$ (high than average but with uncertainty) and visualAbility $0.92 \pm 0.44$ (low), with some a high level of random noise affecting performance $0.48 \pm 0.20$. Finally, '41Animals' with 87% accuracy, shows relatively high

values and low deviations for both navigationAbility $4.37 \pm 0.59$ and visualAbility $1.81 \pm 0.11$, with very little noise $0.05 \pm 0.04$ despite having some non-conformant behaviour in the diagonal. In this case the left-right bias is higher $(0.45 \pm 0.91)$ and may account for this. In general, for those agents where accuracy is too low or too high, the differences are more subtle. Of course, the interesting part of Fig. 23 is the agents in the middle. This suggests that estimating capabilities can benefit from a more focused analysis of instances around the demand thresholds. For 'Juohmaru', this would amount to selecting instances by difficulty, instead of simply selecting all instances for analysis.

# G   SENSITIVITY ANALYSIS

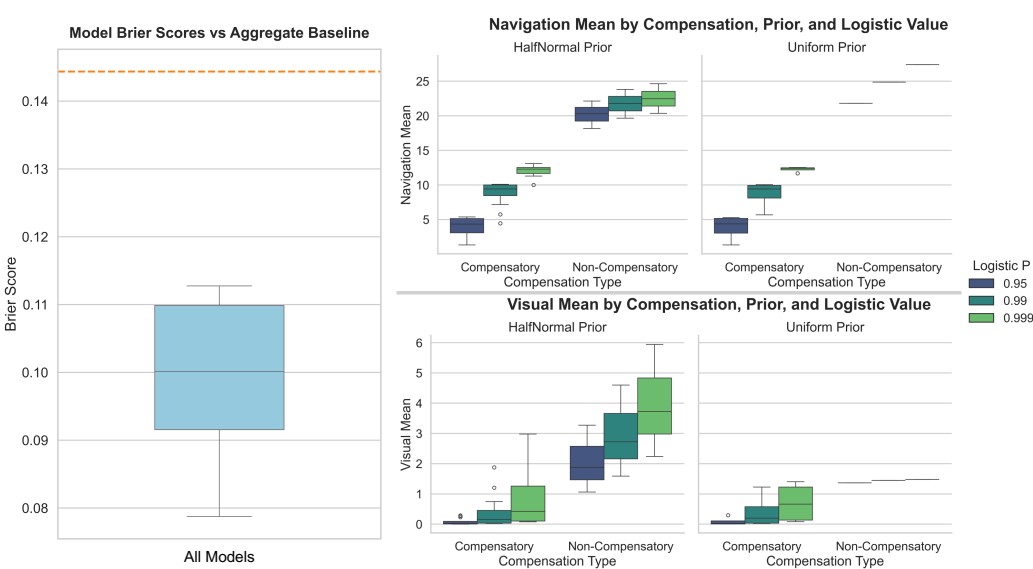

Figure 27: Box plots showing the variation between 60 different measurement layouts with systematically varied structures. Left: The distribution of Brier Scores on a held-out test set, compared to the aggregate score baseline. All models are more predictive than the aggregate. Right: The distributions of estimated mean Navigation Ability (top) and Visual Acuity (bottom), aggregated by compensation, logistic shape, and prior.

To explore the sensitivity of measurement layouts to different network specifications, we conducted a sensitivity analysis, following standard practice in econometrics and cognitive science Saltelli et al. (2008); Steegen et al. (2016). We varied:

- The steepness of the logistic functions, such that the maximum margin produces a probability of 0.95, 0.99, or 0.999.

- The prior: either HalfNormal (with $\sigma = \{1, \frac{1}{2}, \frac{1}{4}\} \times$ max. demand) or Uniform between the minimum demand and the maximum demand plus 20%.

- The relationship between the $n$ capabilities, computing the generalised mean with $p = \{0, 1, 2, 3\}$, with the non-compensatory relationship being the special case of the geometric mean ($p = 0$) raised to the power $n$. All other means implement compensatory relationships.

We fitted every combination of these parameters (60 models) on a single agent from Experiment 1 (Section 3) which had $20 \times 20$ pixel input and 0.1 navigation noise. Figure 27 shows the Brier Scores and estimated means, aggregated across parameter settings. We observe that all models are more predictive than the aggregate, suggesting that these modelling choices to not significantly affect how predictive the measurement layouts are. The estimated mean Navigation Ability and Visual Acuity varies to some degree between parameter settings. Non-compensatory relationships shift capability estimates higher, because the per-capability marginal probabilities are multiplied and therefore need to be higher to obtain high probabilities of overall success. We also notice

that steeper logistic functions push capability estimates higher, particularly for prior with no upper bound (HalfNormal). This is because there is a wider range of capability estimates that give similar probabilities of success, since the logistic is only discriminatory at its inflection point.

These results suggest that modelling choices do not affect how predictive a measurement layout is, but they can change the precise capability measurements. Therefore, it is imperative that users incorporate as much domain knowledge as possible into designing these measurement layouts, for instance, specifying whether and how capabilities are compensatory and whether there are motivated bounds on capability priors.

## H  CODE AND DATASETS

Our experiments did not require a large amount of compute, We ran all of our experiments on Google Colaboratory (Google) using the free tier of service. All corresponding code, datasets and raw results will be released on publication to ensure ease of replication.

