# OpenReview forum: "Inferring Capabilities from Task Performance with Bayesian Triangulation"
_ICLR.cc/2026/Conference — Submitted to ICLR 2026_

### Official Review · Reviewer_pPGY · 2025-10-29

**Soundness:** 3
**Presentation:** 2
**Contribution:** 2
**Rating:** 4
**Confidence:** 2

**Summary:**

This paper presents cognitive profiles that explain behavior in a more fine-grained manner than traditional aggregate benchmark statistics. The authors demonstrate that these profiles lead to improved prediction for performance on new, held-out tasks in synthetic agent experiments.

However, the method is difficult to scale and relies heavily on domain knowledge. While the construction of an effective measurement layout for a real-world task would be a significant contribution, the experiment with real data did not show strong improvements in prediction. Therefore, I recommend against acceptance.

**Strengths:**

The paper provides an intuitive introduction to measurement layouts. The problem is well-motivated, addressing an important challenge in understanding AI system capabilities in a more fine-grained and generalizable way.

For the synthetic agents, the method does appear to improve prediction over aggregate statistics (Figure 5), demonstrating the value of the approach in controlled settings.

**Weaknesses:**

The predictive evaluation results are not very strong or consistent when applying the measurement layout to real data (Figure 8). Given that measurement layouts require domain knowledge and careful effort to construct, the primary contribution should be based on how much they can improve prediction in practical, real-world settings.

The authors construct measurement layouts for a few tasks, but the scalability of this approach remains unclear. While they discuss some general principles in Section 6, there is no systematic procedure for constructing measurement layouts. The bottleneck appears to be domain expertise, which limits the method's practical deployment.

Robustness levels are mentioned as a core component of cognitive profiles in Section 2, but they don't seem to be mentioned later in Section 6 about how to construct measurement layouts. This inconsistency raises questions about the completeness of the framework.

**Questions:**

**Questions for clarification:**

1. (Line 106) "Bias values represent some other preferences or limitations that may impact performance in a less monotonic way." Can you elaborate on how biases are less monotonic than capability levels? In Section 6, they seem to be treated similarly.
2. What led to the choice of linking functions used? Are these chosen based on domain knowledge?
3. (Line 128) "A key conceptual difference between our measurement layouts and typical uses for HBNs is that our approach is trying to capture a hierarchical dependency relation on capabilities and demands, rather than encoding information on hyper-priors." Can you give a typical example use case for HBNs and a use case that measurement layouts would be advantageous for?

---

**Minor comments:**

- Line 19 refers to "cognitive profiles of hand-crafted behavioural agent" while line 104 defines "cognitive profile of an AI system" as a tuple of capability levels, bias, and robustness values. Switching between "system" and "agent" is confusing since "system" can be confused with the agent and the environment combined.
- (Lines 42-44) "We could now infer, by triangulation, that the system's failure is likely a failure of object permanence, a robust inferential method in formal epistemology (Heesen et al., 2019) and causal (Bayesian) reasoning." Please add a citation for triangulation in causal reasoning and consider splitting this sentence for clarity.
- "To achieve this, the team was split into Team A who designed the synthetic agents, and Team B who built the measurement layout without knowledge of the agents built by Team A." Clarify in the text that the team refers to the authors of the paper. While this information is mentioned in the appendix, it should be clear in the main text.
- Maintain consistent x-axis label formatting between Figures 5b, 6, and 9.
- In figure captions or the Qualitative Evaluation paragraph, define what the metrics are for each capability shown in the figures.

---

**Suggestions for improvement (not factored into score):**

1. It would be interesting to conduct some analysis on the evaluation instances in Figure 5a that had low predictive ability. Do these instances correspond to the "fraudster" agents mentioned in the previous paragraph?

---

> ### Author Response · Authors · 2025-11-21
>
> Thank you for your comments and engagement with our paper. We have organised our response according to the weaknesses [W] and questions [Q] you have raised.
>
> ## [W1] Strength of Evaluation Results
>
> The predictive power of the evaluation results when applied to real world data could be improved with, e.g., black box optimisation methods. A good analogy might be that we can create black-box models that can almost perfectly predict human behaviour, but that do not explain why humans behave that way (see Binz et al., 2025, A foundation model to predict and capture human cognition). However,  we believe that a core advantage of the measurement layout approach is that we have good predictive power, while retaining strong explainability and interpretability of the results.
>
> ## [Q1] Bias and Monotonicity
>
> With capabilities it tends to be the case that more is better. That is, more navigation capability generally means that the system will be more likely to succeed at tasks requiring navigation. Biases on the other hand, can be different – for example, in our left-right bias node in experiments 2 and 3, the further away from 0 the value becomes, the less likely the model succeeds. With biases there is a “sweet spot” (usually 0) where deviation from the sweet spot reduces performance. This is what we mean when we say that biases are less monotonic than capabilities. We have updated section 2 to make this more clear.
>
> ## [Q2] Linking Function Choice
>
> The linking functions are based on principles from Item Response Theory, where the sigmoid of (ability - difficulty) is used to model likelihood of success. We have built on this, as well as incorporating domain knowledge (e.g., goals further away are harder to successfully navigate to) to build a cohesive measurement structure for the tasks in question.
>
> Nonetheless, the user of a measurement layout still has significant degrees of freedom when making particular modelling choices, such as the precise shape of the sigmoid link. We now include a sensitivity analysis (as commonly done in econometrics and psychology; Saltelli et al., 2008; Steegen et al., 2016) based on Experiment 1, varying the steepness of the logistic link, the interaction structure between capabilities (non-compensatory vs. several compensatory forms), the prior distribution (Half-Normal vs. Uniform), and the variance of the Half-Normal prior.
>
> All tested models remain more predictive than the aggregate, indicating that these modelling decisions do not substantially change how predictive the measurement layouts are. However, the estimated Navigation Ability and Visual Acuity values do shift somewhat across settings: multiplicative (non-compensatory) relationships and steeper logistic functions tend to produce higher capability estimates, especially under unbounded priors.
>
> Overall, modelling choices affect the exact capability estimates but not the predictive power of the layouts. This highlights the importance of incorporating appropriate domain assumptions when specifying compensatory structure and prior bounds. We have included a new set of figures and a discussion of this sensitivity analysis to Appendix G.

---

> ### Author Response · Authors · 2025-11-21
>
> ## [Q3] Other HBNs
>
> One well known alternative HBN method is HiBayes. Both HiBayES and our Bayesian triangulation framework address different layers of the evaluation problem. HiBayES is a general hierarchical Bayesian GLM framework for evaluation statistics: it treats evaluation data as nested Bernoulli trials and provides model comparison for performance metrics across that hierarchy. In other words, it tells you how to estimate and compare scores in a statistically robust way, given a fixed benchmark structure.
> Our measurement layouts, by contrast, are measurement models in the psychometric sense. They explicitly link instance-level meta-features (e.g., distance, size,
> etc) to latent cognitive capabilities (object permanence, navigation, etc.) via a structured Bayesian network. Bayesian triangulation here means that we infer capability profiles from the pattern of successes and failures across many instances whose meta-features are deliberately varied, rather than from benchmark labels alone. This is what we meant by “capability-oriented inference from instance-level meta-features”: the latent variables are interpretable capabilities, and the covariates are task demands, not just domain or benchmark IDs.
>
> The significance of this difference is that it changes the target of inference: HiBayes yields calibrated estimates of performance with uncertainty at different levels of an existing hierarchy (per domain, per benchmark, per model), whereas measurement layouts yield interpretable cognitive profiles that aim to explain why an agent behaves as it does, in terms of specific strengths and weaknesses. We believe this additional explanatory power of the measurement layouts is a valuable tool for agent evaluation.
>
>
>
>
> Thank you additionally for the minor comments – we have addressed these in the revised submission.
>
> We would like to thank you once again for your comments, which we think have significantly improved the scope, clarity, and interpretation of our paper.
>
>
> ## References
>
> Saltelli, A., Ratto, M., Andres, T., Campolongo, F., Cariboni, J., Gatelli, D., ... & Tarantola, S. (2008). Global sensitivity analysis: the primer. John Wiley & Sons.
>
> Steegen, S., Tuerlinckx, F., Gelman, A., & Vanpaemel, W. (2016). Increasing transparency through a multiverse analysis. Perspectives on Psychological Science, 11(5), 702-712.

---

> > ### Comment · Reviewer_pPGY · 2025-11-23
> >
> > I appreciate the authors' detailed response, including clarification on bias vs. monotonicity, motivation behind linking functions, and discussion on other HBNs, which have improved my understanding of the method. With regard to the strength of evaluation results, while measurement layouts are more transparent than black box methods, one primary proposed contribution is that measurement layouts are "significantly more predictive, accurately estimating performance on new, held-out tasks." I'm not entirely convinced that the results support this claim for real world tasks, especially given that the predictive ability of the measurement layout is compared against a simple baseline of the average performance on the train set rather than the mentioned black-box optimization methods. Therefore, I decide to maintain my score.

---

> > > ### Author Response · Authors · 2025-11-26
> > >
> > > Thank you for your reply, and for highlighting the absence of alternative baselines to our measurement layouts. We have now completed a comprehensive comparison with two other predictive baselines: black box logistic regression and structural equation modelling. We fit two independent logistic regressions which take the demands as input variables and predict either success or correct choice. We also design a structural equation model using the lavaan library in R, with a similar structure to our measurement layout (with the same capability nodes linking to the same demands), serving as an alternative, frequentist, implementation to PyMC. Structural equation models (SEMs) in lavaan are significantly less flexible than PyMC models, offering less control over link functions and compensatory relationships between latent capabilities. Being frequentist, they do not provide posterior probability distributions or probabilities interpretable on a trial-by-trial basis, but they fit more quickly. Figure 8 in the updated manuscript shows that logistic regressions are generally more predictive than both our PyMC models and SEMs, but at the cost of explainability and being able to predict multiple response variables and their interaction under one model. SEMs and our measurement layouts behave similarly, with our PyMC models being more predictive on 9 out of 16 prediction tasks. We would like to emphasise that frequentist structural equation models are simply another way of implementing measurement layouts, returning explainable capability estimates and predictive validity.

---

> ### Comment · Reviewer_pPGY · 2025-11-27
>
> I thank the authors for the additional baselines. This does make the results more comprehensive. I have a few additional questions.
>
> > logistic regressions are more predictive ... at the cost of being able to predict multiple response variables and their interaction under one model.
>
> Can the author's clarify what are the response variables that measurement layouts can predict that logistic regression cannot?
>
> Can the authors also respond to the weaknesses mentioned in the original review about lack of a systematic construction framework and robustness levels?

---

> > ### Author Response · Authors · 2025-11-28
> >
> > Thank you for your questions. Regarding the logistic regressions, we mean that a logistic regression can only predict a single response variable at a time. This means that we ran two independent logistic regressions, one for the success response and one for the correct choice response. However, this does not capture the correlation structure between these response variables. Modelling multiple correlated response variables under the same model essentially requires a form of structural equation model, which can be constructed with and without intervening latents.
> >
> > Regarding systematic construction, we apologise for not responding to this in the first instance. We believe that the systematic construction framework must come from domain knowledge. Following Jo and Wilson (2025), meaningful measurements are contingent on a theory of what the constructs are that we want to measure. Thus, domain knowledge about both domain knowledge of the tasks being solved (e.g., navigation tasks) but also about leading theories from the relevant cognitive / psychological literature about how these capabilities are defined and subdivided, and how they relate to other abilities. This can have the effect of biasing the model, to some extent, towards the structures suggested by the literature, however this is an inherent risk in any capability-based measurement attempt (for example, we only believe that a given test measures X or Y ability due to the theories that define and delineate those abilities). To accurately measure a capability, this domain knowledge is required – it’s very difficult to robustly measure a construct without a rigorous understanding of what that construct is and how it works.
> >
> > Regarding robustness levels, we have now amended section 6 to include discussion of robustness levels, thank you for pointing out this absence. We think robustness levels are important to distinguish from bias and capability because they allow us to account for sources of error both around individual capabilities and globally (see also Jo and Wilson, 2025).
> >
> >
> > Jo, N., & Wilson, A. (2025). What Does Your Benchmark Really Measure? A Framework for Robust Inference of AI Capabilities. arXiv preprint arXiv:2509.19590.

---

### Official Review · Reviewer_MtPb · 2025-10-29

**Soundness:** 3
**Presentation:** 3
**Contribution:** 2
**Rating:** 6
**Confidence:** 4

**Summary:**

The paper describes Heirarchical Bayesian Networks, graphs connecting meta-features of task characterisation with the cognitive profile of a system, to predict observed performance of 3D agents in simulated and real environments. Over the course of three experiments of increasing complexity (simple navigation, simulated object permanence tasks, and real agents solving OP tasks), HBN models assign scores to latent capabilities. The resultant models gave better predictions than predictions based on the average performance on the test set. Additionally, the method successfully identifies deliberate "Achilles' heels" of specially-designed agents.

**Strengths:**

Originality
--
This is, to my knowledge, the most thorough application of HBNs to 3D RL agents.

Quality
--
Leans on strong and established software (PyMC, AAI) inspiring confidence in code correctness. The three experiments span a range of complexity, validating the work in simple situations before taking the work to address complex real-world data. The inclusion of human performance is an added bonus. The paper applies HBNs to a battery of environments, with many tasks, agents, and roll-outs.

Clarity
--
The paper is well laid out and language is overall clear. Figures simply explain measurement layouts and clearly visualise the improvement of Brier score compared to the chosen baseline.


Significance
---
The HBN approach descibed successfully identifies deliberate "Achilles' heels" of specially-designed agents.

**Weaknesses:**

One of the stated advantages of this approach is the possibility to introduce domain knowledge. However, the "bitter lesson" suggests that this approach could be outperformed by simpler approaches combined with scaled compute/data. Although this Bayesian approach is arguably more principled, taking expert domain knowledge into account, in my opinion it needs a comparison to a baseline of a simple exploratory / confirmatory GLM / PCA such as those mentioned in the Related Work.

Minor points: in my opinion, paper flow would be improved by signposting that Related Work is located towards the end of the paper (I was expecting it after the introduction, to establish relevance and to emphasise the distinctions between the current work and existing literature).

**Questions:**

Could the authors expand on the advantage that this Bayesian Triangulation method has over existing Bayesian frameworks such as those mentioned in the Related Work? In particular, could the authors say more about the claimed advantage over HiBayES, that their approach emphasises "capability-oriented inference from instance-level meta-features"? What is the claimed significance of this difference?


L104: Unclear what the justification is for the <C. B. R> tuple. Why this, not something else?

L130: What does it mean in practice to be "trying to capture a hierarchical dependency relation on capabilities and demands, rather than encoding information on hyper-priors"? Isn't this undermined by the predetermination of the Measurement Layout?

Fig 2C: Why did no agent pass more than 50% of the instances? Would you expect the relative performance of model and aggregate to continue past 50% for more capable models? Why or why not?

L184: What are the "random stationary actions" ?

L374: What does it mean to "use the meta-features of each task instance to ensure demands are objective properties of the task"?

In Appendix A, point 3: to what extent does the introduction of general linking functions derived from domain knowledge bias discovered latents towards researcher priors and away from novel discoveries? Would even exploratory GLMs be more appropriate in cases where the number of latent capabilties is unknown? At a fundamental level, doesn't the structure of the measurement layout introduce severe bias, completely determining the latents which can be measured and found?

Perhaps this is what is meant around L394, but I found the language hard to follow at that point.

---

> ### Author Response · Authors · 2025-11-21
>
> Thank you for your comments and engagement with our paper. We have organised our response according to the weaknesses [W] and questions [Q] you have raised.
>
> ## [W2] Signposting Related Work
>
> We have added a short signposting to the introduction that will help readers identify that the related work is towards the end of the paper.
>
> ## [W1/Q1] Comparison to Other (Bayesian) Methods
>
> Regarding comparison to other methods, such as Factor Analysis, Item Response Theory, and Structural Equation Models, we would first like to note that the proper application of these methods requires sampling systems from a well-defined population. Our measurement layout approach is intentionally non-populational because, in an AI evaluation context, it is (a) not clear what the population is that we should sample from: different architectures, training curricula, and hyperparameters may produce i.i.d. populations, a key assumption of these other methods; and (b) there is not often a large enough population to sample from, even if it can be defined. In many contexts, we have access only to a small number of models.
>
> We are currently working on some of the comparisons you suggest, and we will update you as soon as they are ready.
>
>
>
> With regards to HiBayes:
> HiBayES and our Bayesian triangulation framework address different layers of the evaluation problem. HiBayES is a general hierarchical Bayesian GLM framework for evaluation statistics: it treats evaluation data as nested Bernoulli trials and provides model comparison for performance metrics across that hierarchy. In other words, it tells you how to estimate and compare scores in a statistically robust way, given a fixed benchmark structure.
> Our measurement layouts, by contrast, are measurement models in the psychometric sense. They explicitly link instance-level meta-features (e.g., distance, size,
> etc) to latent cognitive capabilities (object permanence, navigation, etc.) via a structured Bayesian network. Bayesian triangulation here means that we infer capability profiles from the pattern of successes and failures across many instances whose meta-features are deliberately varied, rather than from benchmark labels alone. This is what we meant by “capability-oriented inference from instance-level meta-features”: the latent variables are interpretable capabilities, and the covariates are task demands, not just domain or benchmark IDs.
> The significance of this difference is that it changes the target of inference: HiBayes yields calibrated estimates of performance with uncertainty at different levels of an existing hierarchy (per domain, per benchmark, per model), whereas measurement layouts yield interpretable cognitive profiles that aim to explain why an agent behaves as it does, in terms of specific strengths and weaknesses. We believe this additional explanatory power of the measurement layouts is a valuable tool for agent evaluation.
>
> ## [Q2] <C, B, R> Notation
>
> We use the <C,B,R> as a purely organisational tool to help evaluators keep track of the different latent variables that we are inferring. From the HBN’s perspective, the tuple isn’t used – however, we found it helpful to split the responses up when discussing the various elements of the latent variables. We have added a short note detailing this to the paper.
>
> ## [Q3] Hierarchical Dependence
>
> By “"trying to capture a hierarchical dependency relation on capabilities and demands, rather than encoding information on hyper-priors”, we mean that the structure of the measurement layout–when defined by the evaluator– should be built in such a way that captures the relationship between capabilities and demands and how these together affect performance. That is, the evaluator is trying to capture these hierarchy dependencies and relationships. We have rephrased this in the paper to make it more clear.
>
> ## [Q4] Agent Success Rates in Experiment 1
>
> With regards to why no agent passed more than 50% of the instances, we’re assuming you mean in experiment 1 as in the other two experiments some agents do pass more than 50% of the instances. In experiment 1, this is simply because the maximum pixel size made available to the agents was still only a resolution of 40x40 pixels, this makes accurately completing the tasks in the allotted time hard. We expect that with more pixels the agents would perform arbitrarily better, and that model and aggregate performance would mirror the LHS of the figure. We limited this experiment to just 100 agents (10 pixel size levels by 10 noise levels), since we also run each of them on 100 trials, thus requiring 10,000 instances of Animal-AI to complete.

---

> ### Author Response · Authors · 2025-11-21
>
> ## [Q5] Random Actions
>
> Random stationary actions refer to the random chance that an agent produces a stationary action (i.e., stays still) at a particular time step  – thus making it harder to complete the task. The probability of a random stationary action is controlled by the Navigation Noise task demand.
> We have slightly rephrased this to be more explicit and easy to follow.
>
> ## [Q6] Meta-feature demands
>
> With regards to what  "use the meta-features of each task instance to ensure demands are objective properties of the task" means. We mean that we ensure that demands are observable properties of the environment — meta-features should be very easy to measure, be objective measurements, and therefore easy to identify when building test batteries. We have rephrased this for clarity in the paper.
>
> ## [Q7] Role of Domain Knowledge
>
> The measurement layouts are intended to be constructed from domain knowledge. This includes both domain knowledge of the tasks being solved (e.g., navigation tasks) but also about leading theories from the relevant cognitive / psychological literature about how these capabilities are defined and subdivided, and how they relate to other abilities. This can have the effect of biasing the model, to some extent, towards the structures suggested by the literature, however this is an inherent risk in any capability-based measurement attempt (for example, we only believe that a given test measures X or Y ability due to the theories that define and delineate those abilities). To accurately measure a capability, this domain knowledge is required – it’s very difficult to robustly measure a construct without a rigorous understanding of what that construct is and how it works.
>
> We would like to thank you once again for your comments, which we think have significantly improved the scope, clarity, and interpretation of our paper.

---

> > ### Author Response · Authors · 2025-11-26
> >
> > We would like to update the reviewer that we have also been able to complete a comparison with a black-box prediction model (logistic regression) and a confirmatory factor analysis (a structural equation model), in response to their comment.
> >
> > We fit two independent logistic regressions which take the demands as input variables and predict either success or correct choice. We also design a structural equation model using the lavaan library in R, with a similar structure to our measurement layout (with the same capability nodes linking to the same demands), serving as an alternative, frequentist, implementation to PyMC. Structural equation models (SEMs) in lavaan are significantly less flexible than PyMC models, offering less control over link functions and compensatory relationships between latent capabilities. Being frequentist, they do not provide posterior probability distributions or probabilities interpretable on a trial-by-trial basis, but they fit more quickly.
> >
> > Figure 8 in the updated manuscript shows that logistic regressions are generally more predictive than both our PyMC models and SEMs, but at the cost of explainability and being able to predict multiple response variables and their interaction under one model. SEMs and our measurement layouts behave similarly, with our PyMC models being more predictive on 9 out of 16 prediction tasks. We would like to emphasise that frequentist structural equation models are simply another way of implementing measurement layouts, returning explainable capability estimates and predictive validity.

---

### Official Review · Reviewer_2vhw · 2025-10-31

**Soundness:** 2
**Presentation:** 3
**Contribution:** 2
**Rating:** 2
**Confidence:** 3

**Summary:**

The paper proposes measurement layout as an alternative to aggregate statistics for evaluating ML models.

**Strengths:**

- The paper is mostly well-written and easy to follow.
- Measurement layouts provide an interesting means to evaluate ML models on static benchmarks (and the experiments demonstrate that).

**Weaknesses:**

- The major limitation of this work is that the measurement layout has to be manually designed by the humans.
- The population of agents used to fit the measurement layouts seems to be created in an ad-hoc manner.

**Questions:**

1. Can the measurements layouts be automatically generated, without human intervention?
2. Is there a systematic way to generate the population of agents used to fit these layouts?

---

> ### Author Response · Authors · 2025-11-21
>
> Thank you for taking the time to read and engage with our paper.
>
> ## [Q1] – Manual Specification of the Measurement Layouts
> We agree that our current layouts are manually specified. While this increases the effort required to make use of measurement layouts, it also comes with positives, and we believe requires a more nuanced analysis. First, it keeps the evaluation model interpretable and auditable: every node and edge in a layout has a clear meaning (e.g., “time under occlusion loads on object permanence”), rather than being the result of an opaque optimisation. Second, it lets us directly encode domain expertise and theory-driven constructs about capabilities and task demands, in the same way that psychometrics and cognitive science batteries are designed around hypothesised abilities such as working memory or object permanence. In many of the settings we care about (e.g., comparing RL agents to children on object permanence tasks), the choice of which capabilities to measure and how task features relate to them is intrinsically normative and theoretical, and we do not think this should be delegated entirely to an automatic procedure. We have added a discussion on the trade-offs of automated vs manual evaluation to Section 8 of the paper.
>
> ## [Q2] Agent population
> On the population of agents: our core use case does not rely on populations of agents. Measurement layouts are fitted to a single agent/system, and we use populations mainly to (i) test recovery when ground-truth capabilities are known (Experiments 1–2), and (ii) illustrate how different cognitive profiles (e.g., PPO vs Dreamer vs children) occupy different regions of the same layout (Experiment 3). In Experiments 1–2, the populations are not arbitrary; they are constructed to cover the capability space (including “Achilles heel’’ agents with targeted weaknesses) so that we can assess recovery and predictive validity across a broad range of profiles. We have updated Sections 3 and 4 to make this clearer.
>
>
> We would like to thank you once again for your comments, which we think have significantly improved the scope, clarity, and interpretation of our paper.

---

### Official Review · Reviewer_Cxzo · 2025-11-03

**Soundness:** 3
**Presentation:** 2
**Contribution:** 2
**Rating:** 2
**Confidence:** 2

**Summary:**

This paper proposes measurement layouts, hierarchical Bayesian networks that infer cognitive profiles (capabilities, biases, robustness) of AI systems from performance patterns across diverse tasks. The core contribution is Bayesian triangulation, leveraging task-instance features that vary independently to disambiguate which capabilities drive success or failure. The work is technically sound and addresses a genuine problem in AI evaluation related to moving beyond aggregated benchmark statistics to interpretable, capability-level profiles. The three experiments demonstrate proof-of-concept with increasing complexity, and the approach shows promise in predicting performance on held-out tasks.

However, the paper oversells the generality of its framework without adequately addressing fundamental identifiability constraints. Most critically, the central claim about "triangulation" relies on an implicit assumption that rarely goes unchallenged: that you can always construct enough independent task dimensions to triangulate the capabilities of interest. This might not be feasible in many tasks,

**Strengths:**

1. Applying psychometric ideas to AI evaluation is creative and timely.

2. Three increasingly complex scenarios provide scope for this work.

**Weaknesses:**

1. This is your key insight, and it's a real problem. The paper invokes "triangulation" as though it's a general principle, but the method fundamentally requires that task dimensions are sufficiently independent and align with capability structure.
The core issue: In Experiment 1, you have 2 task demands (goal size, goal distance) and 2 capabilities (visual acuity, navigation). That's nearly square which is barely enough degrees of freedom. In Experiment 2, you have 12 task dimensions but 7 capabilities. Yet the paper never formally addresses: when does the system become underdetermined?

2,  If you have $k$ capabilities and $d$ task dimensions, when is the system identifiable? The paper has no theorem or even heuristic guidance. For instance, if navigation performance depends on "both" visual acuity and memory, but your task dimensions only vary goal distance and visibility, can you separate these effects?

3. The paper acknowledges (Appendix C.2) that some task dimensions were not included in the measurement layout because Team B didn't realize they existed (e.g., occluder RGB values). This is presented as a strength by showing robustness to model misspecification, but it actually undermines the triangulation premise. If you omit a task dimension that affects performance, you risk confounding it with capability estimates. The paper should quantify this risk.

4. The paper claims the approach is "agnostic to the task or type of system" (lines 84-87), yet constructing measurement layouts requires substantial expertise.

5. Fitting profiles for DRL agents and children is impressive, but the evaluation is thin. There is no greound truth. Unlike Experiment 2, you don't know the true capabilities of PPO or Dreamer agents. Showing that you can fit a model and make predictions doesn't validate that you've recovered anything meaningful.

6. The paper cites psychometric principles from IRT and SEM but doesn't formalize the connection. You invoke "hierarchical dependency relation on capabilities and demands" (line 128-129) but never define this formally. What's the mathematical structure of your HBN beyond the linking functions?

**Questions:**

1. Can you formalize the identifiability condition? Under what constraints on the number of capabilities $k$, task dimensions dd
$d$, and linking function structure is the posterior over cognitive profiles unique (up to label swapping)?

Why not validate measurement layouts against ground truth beyond Experiment 2? For real agents, can you compare inferred capabilities against interpretability methods (e.g., saliency maps, attention weights, or probing classifiers)?
How sensitive are results to linking function choice? Can you show that products vs. minimums vs. generalized means give consistent capability orderings?

---

> ### Author Response · Authors · 2025-11-21
>
> Thank you for your comments and engagement with our paper. We have organised our response according to the weaknesses [W] and questions [Q] you have raised
>
> ## [W1/W2/Q1.1] Capability Identifiability
>
> We agree that triangulation is only possible when tasks have been appropriately designed, and we now make this explicit in the text. Our framework does not assume that capabilities are always identifiable; instead, identifiability depends on how task demands vary across instances.
>
> Concretely, for each task instance we record observable meta-features (e.g., distance, size, time under occlusion). For each capability k, we define a simple scalar effective demand $d_i,k=h_k(d_i)$  from these meta-features (often just selecting the relevant feature or an indicator, e.g. goal distance for navigation, presence of lava for lava-ability). Collecting these values into a matrix with entries $X_ik=d_i,k$  gives a design matrix: rows are instances, columns are capabilities. In our non-compensatory layouts, each capability compares its ability c_k​ to its demand via a margin $c_k−d_i,k$, which is mapped to a per-capability success probability $q_i,k=\sigma(c_k−d_i,k)$. The overall success probability on instance i is then $p_i(c)= \prod_k q_i,k$.
>
> Identifiability of the capability vector $c$ does not depend on simply counting “dimensions’” versus “capabilities’’; it depends on the patterns in this design matrix. Intuitively, triangulation is possible when (i) each capability is differentially tested by certain instances (i.e., some instances mostly test object permanence and not navigation, while others show the inverse pattern) , so performance change can be attributed to that capability, and (ii) different capabilities are tested in different ways across instances, rather than demands being strongly correlated across instances. If two capabilities are always co-engaged in exactly the same pattern across tasks, then no model—ours or any other—can truly separate them, since this amounts to perfect collinearity between the predictor variables. Indeed, minimizing this collinearity is a key criterion for effective task design in the psychometric measurement setting.
>
> In our experiments we deliberately design the tasks to satisfy these conditions. In Experiment 1, there are instances where navigation is trivial but visual acuity varies and vice versa, so distance and size do not change in lockstep. In Experiment 2, we include OP-free controls as well as ramp-only and lava-only conditions alongside combinations that jointly test multiple abilities, so that each capability is sometimes tested in isolation and sometimes appears in distinct combinations. We will add a short subsection clarifying these design requirements.
>
> ## [W3] Missing Task Dimensions e.g., Colours
>
> We agree that missing key demands or capabilities can lead to non-optimal inference of capabilities or prediction of performance. However, it is important to keep in mind that an inherent problem with any top-down, expertise driven structure is that experts do not know what they do not know. However this does not undermine the value of having such input, but merely emphasises the importance of evaluation and modelling as an iterative process, where additional details need to be identified and incorporated into the evaluation until a satisfactory and sufficient level of predictability is found.
>
> ## [W4] System/Task Agnosticism
>
> By “agnostic to the task or type of system" we mean that the measurement layouts approach can be applied to any type of decision making system – e.g., RL agents, LLMs, other ML techniques, and even humans. This is orthogonal to how much domain knowledge is required to build the measurement layouts. That said, we believe the domain knowledge requirement is more nuanced than a wholly negative factor:
>
> First, it keeps the evaluation model interpretable and auditable: every node and edge in a layout has a clear meaning (e.g., “time under occlusion loads on object permanence”), rather than being the result of an opaque optimisation by, say, a neural network assessor model. Second, it lets us directly encode domain expertise and theory-driven constructs about capabilities and task demands, in the same way that psychometric and cognitive science test batteries are designed around hypothesised abilities such as working memory or object permanence. In many of the settings we care about (e.g., comparing RL agents to children on object permanence tasks), the choice of which capabilities to measure and how task features relate to them is intrinsically normative and theoretical, and we do not think this should be delegated entirely to an automatic procedure. We have added a discussion on the trade-offs of automated vs manual evaluation to the paper.

---

> ### Author Response · Authors · 2025-11-21
>
> ## [W5/Q1.2] Ground Truth in Experiment 3
>
> The reviewer is correct to point out that there is no ground truth in Experiment 3, at least for the deep RL agents. We believe this is an important test case because in AI Evaluation, we are usually trying to measure the capabilities of systems for which we do not know the ground truth. The properties we are trying to measure – visual acuity, navigation ability etc. are abstract constructs that can only be indirectly measured — through techniques such as the measurement layouts we are proposing. Abstract constructs are notoriously difficult to measure accurately and so having a ground truth is extremely non-trivial:  If we had a robust way of getting the ground truth for these tasks we wouldn’t need the measurement layouts approach.
>
> Using a standard approach in measurement theory and psychometrics, we validate our measures by checking how well they predicted the performance data of the agents on held out task instances – predictive validation is a standard approach to validating indirect measurements within measurement theory. In Experiment 3, we also use three classes of agents which have ground truth capability profiles derived from theory: we designed the heuristic agent to be good at navigation but bad at object permanence; and we know (from psychology) that human children over the age of 4 years old have object permanence and are broadly capable of playing videogames. We can use these facts to calibrate our measurement layouts and make sure that our extracted capability profiles broadly cohere with what we would expect from those agents. In this way, we can have some degree of trust in the measurements for the DRL agents without necessarily having a ground truth for them. We have added a few sentences to this effect in Section 5 of the paper.
>
> ## [W6] Mathematical Structure of HBNs
>
> The full mathematical structure of the HBNs used in our work are presented in Appendix D, where the linking functions, and prior distributions of all nodes are provided.
>
> ## [Q1.3] Interpretability Methods
>
> It’s unclear what other benefits the suggested interpretability measures would provide in terms of validating the accuracy of a measurement in the context of these RL systems. As far as we are aware, interpretability methods are typically used to explain behaviour according to model internals – but these do not track onto measuring abstract capabilities in a meaningful way. We have added a short explanation to the paper explaining the significance of predictive validation for our experiments and measures.
>
> ## [Q1.4] Model design sensitivity
>
> Thank you for pointing out the need to check the robustness of our results to modelling assumptions. We now include a sensitivity analysis (as commonly done in econometrics and psychology; Saltelli et al., 2008; Steegen et al., 2016) based on Experiment 1, varying the steepness of the logistic link, the interaction structure between capabilities (non-compensatory vs. several compensatory forms), the prior distribution (Half-Normal vs. Uniform), and the variance of the Half-Normal prior.
>
> All tested models remain more predictive than the aggregate, indicating that these modelling decisions do not substantially change how predictive the measurement layouts are. However, the estimated Navigation Ability and Visual Acuity values do shift somewhat across settings: multiplicative (non-compensatory) relationships and steeper logistic functions tend to produce higher capability estimates, especially under unbounded priors.
>
> Overall, modelling choices affect the exact capability estimates but not the predictive power of the layouts. This highlights the importance of incorporating appropriate domain assumptions when specifying compensatory structure and prior bounds. We have included a new set of figures and a discussion of this sensitivity analysis to Appendix G.
>
>
>
> We would like to thank you once again for your comments, which we think have significantly improved the scope, clarity, and interpretation of our paper.
>
>
> ## References
>
> Saltelli, A., Ratto, M., Andres, T., Campolongo, F., Cariboni, J., Gatelli, D., ... & Tarantola, S. (2008). Global sensitivity analysis: the primer. John Wiley & Sons.
>
> Steegen, S., Tuerlinckx, F., Gelman, A., & Vanpaemel, W. (2016). Increasing transparency through a multiverse analysis. Perspectives on Psychological Science, 11(5), 702-712.

---

### Author Response · Authors · 2025-11-21

Dear Reviewers and Area Chair,

We appreciate the reviewers taking time to engage with our work and want to thank them for their active participation in the review process. We are happy to hear that the reviewers found our paper to be “technically sound” (`Cxzo`), “well-motivated” (`pPGY`) and addressing “a genuine problem in AI Evaluation” (`Cxzo`, `pPGY`), using “strong and established software” (`MtPb`).

In our individual responses to each reviewer, we tried to address their questions and concerns. Over the course of the rebuttal, we have added the following:

Further detail on how measurement layouts relate to other approaches such as HiBayES (`MtPb`)
A sensitivity analysis to determine how sensitive measurement layouts are to different modelling choices (`Cxzo`, `pPGY`).
A detailed account for how domain expertise should interact with measurement and evaluation practice in AI (`Cxzo`, `2vhw`, `pPGY`)
Overall improvements to the readability and clarity of the paper, thanks to the detailed comments of all reviewers.

We again want to thank our reviewers for their time and their productive comments, which we are confident have improved the scope and clarity of our results.

---

> ### Author Response · Authors · 2025-11-26
>
> Dear Reviewers and Area Chair,
>
> In response to the comments, we have also been able to complete a further revision to our manuscript. We now include a complete comparison with two other predictive baselines: black box logistic regression and structural equation modelling.
>
> We fit two independent logistic regressions which take the demands as input variables and predict either success or correct choice. We also design a structural equation model using the lavaan library in R, with a similar structure to our measurement layout (with the same capability nodes linking to the same demands), serving as an alternative, frequentist, implementation to PyMC. Structural equation models (SEMs) in lavaan are significantly less flexible than PyMC models, offering less control over link functions and compensatory relationships between latent capabilities. Being frequentist, they do not provide posterior probability distributions or probabilities interpretable on a trial-by-trial basis, but they fit more quickly.
>
> Figure 8 in the updated manuscript shows that logistic regressions are generally more predictive than both our PyMC models and SEMs, but at the cost of explainability and being able to predict multiple response variables and their interaction under one model. SEMs and our measurement layouts behave similarly, with our PyMC models being more predictive on 9 out of 16 prediction tasks. We would like to emphasise that frequentist structural equation models are simply another way of implementing measurement layouts, albeit with less flexibility than HBNs in PyMC - both return explainable capability estimates and enable predictions about new unseen instances.

---

### Meta-Review · Area_Chair_37A7 · 2026-01-04

**Summary:**

The paper proposes Bayesian triangulation for measurement layouts, a new method for evaluating AI agents by identifying the connections between their capabilities and the properties of different tasks. The reviewers acknowledged that the paper targets a very important problem in AI evaluation, looks sound, and is novel. However, they also pointed out several weaknesses, the most prominent of which are:

(1) The need to specify measurement layouts manually, the feasibility and scalability of which for realistic agents was called in question by several reviewers. The authors didn't directly challenge the claim that constructing these layouts can be difficult but instead pointed out the method's advantages over the alternatives:  interpretability and strong predictive power of the resulting analysis.

(2) The limitations of triangulation at identifying connections between capabilities and task dimensions. The authors acknowledged that these limitations are indeed fundamental but also provided some insights about the factors that can facilitate identifiability.

(3) Lack of sensitivity analysis. The authors have addressed this in the rebuttal stage.

(4) Empirical comparison to other methods. The authors have addressed this as well.

After reading the paper, the metareviewer believes that the method, while interesting, is indeed hard to apply in practice both due to the difficulty of defining a meaningfully measurable task characterization and of constructing cognitive profiles for powerful agents.

For instance, what is a meaningful characterization for the task of accurately summarizing a research paper? What are meaningful but at the same time measurable meta-features for it? Similarly, how do we construct a cognitive profile for an LLM-based agent, a profile that would be relevant to the task of accurate research paper summarization? This is extremely non-trivial, given that GenAI agents have a set of capabilities (and weaknesses) that a user may not understand or be aware of. One example is ability to "comprehend" a figure from tikz code describing the figure. Another example are their difficulty with counting the number of occurrences of a letter in a word, such as the number of "r"s in "strawberry" - this failure doesn't stem from Gen AI agents' (in)ability to count in general, as one might assume, but instead has a technical nature specific to text represented as characters rather than as an image.

Essentially, Section 6, which attempts to answer these questions, is both too high-level and makes the dangerous assumptions that human capabilities are applicable to Gen AI agents and that task demands that influence human performance on a task are meaningful for these agents. The problem of constructing measurement layouts is much more complicated than this article's examples make it seem. Without making progress on it's unclear how to apply the paper's methodology to analyzing Gen AI agents - one of the most important classes of agents in use today. Because of this, at least in the absence of experiments on Gen AI agents, the impact of this paper's contribution appears limited.

**Reviewer Concerns:**

Concerns (3) and (4) above were addressed, but (1) and (2) weren't fully - see the details above.

**Reviewer Scores:**

Only one reviewer, pPGY, responded the rebuttals and indicated their intention to keep their score, 4. The two reviewers who initially gave a score of 2, Cxzo and 2vhw, may have increased their scores but almost surely not above 4, given that the rebuttal didn't really address their main concerns. The remaining reviewer, MtPb, is likely to have kept their score at 6.

---

### Decision · Program_Chairs · 2026-01-26

Reject